# "Gender dimensions of quality of life": The determinants of life satisfaction among South Africans residing in the Gauteng province— Using a multilevel psychodemographic analysis of the GCRO's quality of life survey (2009–2024)

**Monica Ewomazino Akokuwebe**[1]\*, **Salmon Likoko**[2],
**Shamsunisaa Miles-Timotheus**[3], **Moyahabo Mabala**[4]

1 SAMRC Developmental Pathways for Health Research Unit, Department of Paediatrics, School of Clinical Medicine, Faculty of Health Sciences, University of the Witwatersrand, Johannesburg, South Africa, 2 Demography and Population Studies, School of Social Sciences, University of Witwatersrand, Johannesburg, South Africa, 3 Gauteng City-Region Observatory (GCRO), University of the Witwatersrand, University of Johannesburg, Johannesburg, South Africa, 4 Group Finance, Johannesburg, South Africa

\* monica.akokuwebe@wits.ac.za

## Abstract

This study examined gender-related differences in life satisfaction among 70,314 South Africans (33,325 males and 36,989 females) using data from the Gauteng City-Region Observatory (2009–2024). Life satisfaction levels (suffering, struggling, thriving) were analyzed in relation to demographic factors including age, education, employment, income, and population group. Using multilevel psychodemographic analyses, the study uniquely identifies how these factors operate differently for males and females over a 15-year period. Education, employment, and income were significant predictors of life satisfaction (all $p < 0.01$); for males, income had stronger indirect effects, while for females, education ($\beta = 0.40$, $p < 0.01$) and income ($\beta = 0.93$, $p < 0.001$) exerted direct effects. These findings reveal gender-specific pathways to life satisfaction, highlighting the importance of considering both individual and structural determinants. The results have policy relevance by underscoring the need for targeted, gender-sensitive interventions in education, employment, and income support to enhance well-being and promote equity among South Africans in Gauteng province.

## 1. Introduction

Understanding Quality of Life (QoL) is central to evaluating subjective well-being, with life satisfaction recognized as a key socio-economic indicator [1–4]. Psycho-demographic research highlights how demographic characteristics, gender identity,

**Data availability statement:** All relevant data underlying the findings of this study will be made available in a public repository upon acceptance of the manuscript. De-identified data will be deposited in the Journal repository and a DOI link will be provided in the final published article. Any potentially identifying or sensitive participant information will be removed to ensure compliance with ethical approval and participant confidentiality. Further, data are from the Gauteng City-Region Observatory (GCRO) and the dataset is open to qualified researchers free of charge. In order to access the data from the GCRO, a written request was submitted to the GCRO and permission was granted to use the data for this survey. To request access to the dataset, please apply at https://www.datafirst.uct.ac.za/dataportal/index.php/catalog/GCRO/?page=1&sort_by=title&sort_order=asc&ps=15&repo=GCRO Supplementary Information for Data and Materials: All relevant data supporting the findings of this study are included in the published article and its Supporting Information files. The do-files used to analyze the datasets are also provided within the Supporting Information to ensure full reproducibility of the results.

**Funding:** The author(s) received no specific funding for this work.

**Competing interests:** No competing interests.

**Abbreviations:** QoL, Quality of Life; GCRO QoLS, Gauteng City-Region Observatory's Quality of Life Survey; RSA, Republic of South Africa; SALGA, South African Local Government Association; GCR, Gauteng City Region; SWLS, Satisfaction with Life Scale; gSEM, generalized Structural Equation Modelling; HH, household; SASSA, South African Social Security Agency; GCRO, the Gauteng City Region Observatory.

language, socialization, racial group membership, and cultural individuality influence subjective well-being through the interplay of external conditions and internal psychological processes [3–7]. Life satisfaction thus reflects how individuals compare their current circumstances to their ideal life, considering both material and spiritual aspects [8,9]. In South Africa, integrating life satisfaction measures into national policy frameworks could improve monitoring of demographic indicators such as education, employment, income, and access to resources [10,11]. South Africa's multicultural society presents a complex landscape for studying life satisfaction, where social status and persistent inequalities strongly influence well-being [12–17]. Despite social progress, the country remains one of the most unequal globally, with a predicted Gini coefficient of 0.63 in 2024 [11]. Previous studies have shown that while improved living standards enhanced quality of life for Black South Africans, disparities persisted compared to other groups [7]. Economic challenges, unemployment, political changes, and the COVID-19 pandemic have further shaped public trust, governance, and life satisfaction [4,14].

Life satisfaction reflects a multidimensional comparative process that spans happiness, economic stability, social relationships, and well-being [8,12]. Existing research in South Africa has reported mixed findings on the effects of economic and demographic determinants [18,19]. Although numerous theoretical and empirical studies exist, none provide universal explanations [20–23], and gender-based differences remain underexplored. Psychodemographic influences such as education, employment, and income often vary by gender, complicating the task of distinguishing direct effects from social and cultural selection processes [1,2]. Globally, determinants of life satisfaction include age, gender, race, marital status, health, education, and income [14,24]. Similar associations have been reported across sub-Saharan Africa, including Ghana, Tanzania, Ethiopia, and Malawi [1,16]. In South Africa, factors such as religion, migration, social investment, job security, poverty, and class status have also been found to influence well-being [23–27]. Recent South African studies have investigated social networks, migration, demographics, and racial differences [4,8], but methodological limitations remain. Determinants such as chronic illness, poor healthcare, unemployment, and inequality continue to shape life satisfaction outcomes [2,27].

The Gauteng City-Region Observatory (GCRO) Quality of Life Survey (2009–2024) [28–34] provides a unique opportunity to address these gaps. This study uniquely examines life satisfaction in Gauteng using a multilevel psychodemographic approach to a large, longitudinal dataset over a 15-year period (2009–2024), integrating individual, household, and community-level factors while explicitly considering gender differences. Unlike previous research, it combines long-term survey data with multilevel modeling to reveal how socio-economic, cultural, and environmental determinants intersect to shape life satisfaction across sexes. Importantly, it emphasizes gender differences in life satisfaction—a dimension rarely examined systematically in South Africa. Several studies have shown that education, work, and societal factors are more strongly associated with male life satisfaction, while female life satisfaction is linked to marital status and the quality of social relationships [12,18]. It is therefore

expected that our findings will reflect these gendered differences, addressing a gap in existing research. This study examines life satisfaction by sex distribution among South Africans, with specific aims to: (1) measure life satisfaction across sex and age groups; (2) assess bivariate associations between life satisfaction and demographic determinants by sex; and (3) propose a mediation model linking demographic determinants to life satisfaction by sex across survey years.

## 2. Methods

### 2.1. Study area

This study was conducted in Gauteng Province, South Africa's smallest and most urbanized province [35]. South Africa, officially the Republic of South Africa (RSA), has a population of nearly 60 million and borders six countries while surrounding Lesotho. It operates a tripartite capital system: Pretoria (executive), Bloemfontein (judicial), and Cape Town (legislative) [35]. The country is ethnically diverse, with Black South Africans making up over 81% of the population, and Christianity as the predominant religion [35,36]. Despite being an upper-middle-income economy and a member of G20, BRICS, and the Commonwealth, South Africa faces high levels of poverty, unemployment, and inequality, with 25% of its population living on less than US$1.25 per day as of 2008 [36]. A notably youthful demographic, with 37% aged 14–35, characterizes its population profile [36,37]. Gauteng is South Africa's most populous and urbanized province, with 15 million residents in 2022, accounting for 26.6% of the national population [38,39]. It contributes about a third of the country's GDP, producing R1.59 trillion in 2016 and attracting over a third of internal migrants, while provinces like Limpopo and Eastern Cape experienced outflows [35,37,38]. By 2022, 89.7% of households had access to flush or chemical toilets, up from 86.5% in 2011, reflecting infrastructure progress [40–42]. The province's cosmopolitan character is evident in its diverse languages, with isiZulu (25%), Sesotho (13.4%), Sepedi (12%), English (10.7%), and Setswana (9.8%) widely spoken, alongside all eleven official languages [33,35,37,40].

### 2.2. Study design and data source

This study draws on the Gauteng City-Region Observatory (GCRO) Quality of Life (QoL) survey data collected between 2009 and 2024. The GCRO is a partnership between the University of Johannesburg, the University of the Witwatersrand, the Gauteng Provincial Government, and the South African Local Government Association [28]. The QoL survey is a large cross-sectional household survey targeting adults (18+) in Gauteng, designed to capture socioeconomic conditions, perceptions of public services, psychological outlooks, values, and overall quality of life [28–34]. Data support provincial monitoring, Sustainable Development Goals (SDGs) tracking, and policy development [43–45]. The survey employed a multi-stage, stratified cluster sampling approach. Since its inception in 2009, GCRO has completed seven rounds: Round 1 (2009) [28], Round 2 (2011) [29], Round 3 (2013–2014) [30], Round 4 (2015–2016) [31], Round 5 (2017–2018) [32], Round 6 (2020–2021) [33], and Round 7 (2023–2024) [34]. Across these rounds, 86,832 enumeration areas were sampled, with 70,314 individuals forming the final analytic dataset (male = 33,325; female = 36,989) [28–34]. Within municipalities, sampling was stratified by urban–rural distribution and household counts, ensuring representativeness.

The authors accessed anonymized secondary data via the GCRO website on 7 August 2024. No personally identifiable information was available during collection or analysis, as datasets were provided under GCRO's data-sharing protocols.

### 2.3. Study population and sample size

The study population comprised South African nationals stratified by sex who resided in the Gauteng province of South Africa. South African nationals refers to individuals who hold South African citizenship. By conceptual definition and operationalization in this study, a South African national is any person born to at least one South African parent who receives citizenship at birth; children born to a legal resident of the country are permitted to obtain South African citizenship only when they reach the age of majority. Also, foreign nationals may be granted citizenship after meeting a residence

requirement (usually five years). This definition was used in line with the South African nationality law detailing the conditions by which a person is a national of South Africa. The primary law governing such nationality requirements is the South African Citizenship Act 1995, which came into force on the 6th of October 1995 [46]. The total number of South African nationals from the 2009–2024 GCRO datasets was 70,314 comprising 33,325 males and 36,990 females for the sample size for this study, after the exclusion of missing variables in the datasets cleaning. The sample size of 70,314 included the age range from 18 to 48 + years [28–34], and this comes after the datasets from different survey years were combined. The data from each year was combined to create a comprehensive dataset, enabling a broader perspective on trends and changes over time. The datasets from 2009–2024 were added together to expand the range of generalization in order to improve the statistical power of analysis for reliable conclusions, allowing comparative analysis for potential causal relationships and control for confounding variables, hence justifying its inclusion [47–49]. In addition, an iteration variable was created to be able to identify changes over time.

## 2.4. Operational definition of concepts

Sex and gender are two different concepts, but they are often used interchangeably [50,51]. Sex refers to biological differences (such as chromosomes, hormones, and reproductive organs) [50] while gender refers to socially constructed roles, behaviours, activities, and expectations associated with femininity and masculinity [51]. Therefore, in this study, we treated sex and gender as distinct constructs. Sex was used to stratify the population into male and female categories, while gender was utilized to explore roles, identities, and societal expectations related to life satisfaction.

## 2.5. Measures

The GCRO Quality of Life Rounds 1–7 (2009–2024) [28–34] survey—Full Questionnaires were included in the field data collection instrument, which comprised questions focused on: (1) demographic details of the enumerated population (population group, sex, age, language), (2) housing (dwelling type, tenure, satisfaction with dwelling, perceived quality of housing and housing allocation), (3) household services (water, sanitation, refuse removal, energy sources), (4) migration and health (including disability), (5) education and employment (including employment sector), (6) community services (amenities, transport, leisure activities, safety and crime), (7) financial data (including debts, income, and social grants), (8) household assets (Telephone, Television, Computer, Radio, Music system, Satellite TV [e.g., MNET, DSTV], Internet connection, Car, Bicycle, Fridge), (9) public participation and governance, (10) perceived personal well-being, and 11) quality of life of respondents. We used data from questionnaires administered to randomly selected women and men living in the surveyed households [28–34].

## 2.6. Outcome variable

The outcome variable, overall life satisfaction, was measured using the Satisfaction with Life Scale (SWLS), a versatile tool that can be adapted to different cultural contexts. Initially developed by Diener et al. [52], this five-item instrument asks respondents to rate aspects of their lifestyle on a scale from 1 ("very dissatisfied") to 5 ("very satisfied"), with total scores calculated as recommended by Diener et al. [53]. Building on this, Deaton [54] proposed a further evaluation by integrating Cantril's Self-Anchoring Ladder of Life Satisfaction [55] alongside Gallup's measure [56]; in this approach, respondents indicate their life satisfaction on a scale ranging from 0 to 10. This modification was designed to provide a more nuanced exploration of life satisfaction determinants while accounting for various psychodemographic factors. Thus, during the survey, participants reported their satisfaction level and position on the ladder. Following the adoption of Cantril's and Gallup's scales, the SWLS scores were reclassified into three categories: "Thriving" for scores 7–10 (coded as 2), "Struggling" for scores 5–6 (coded as 1), and "Suffering" for scores 0–4 (coded as 0). This recording enhances interpretability and offers a clearer picture of overall well-being in the South African context. Prior studies have validated the SWLS with Cantril's Ladder and Gallup's approach as reliable and robust tools for assessing life satisfaction in various settings [56–60].

 

**2.6.1. Operationalization of life satisfaction levels.** The operational definitions of the levels of life satisfaction (suffering, struggling, and thriving) provide a clear understanding of the respondents' varying levels of life satisfaction and well-being, using Gallup [54] and Cantril's Self-Anchoring Ladder of Life Satisfaction measures [55]. Thus,

- **Suffering (0–4)**: Respondents in this category face significant challenges, including health, social, or economic issues. They often struggle to achieve satisfaction in their life, or maintain a stable quality of life, leading to poor overall well-being [52–55].

- **Struggling (5–6)**: Respondents in this category are moderately satisfied with their lives. Despite occasional health challenges, they manage life's obstacles, alternating between periods of stability and hardship [52–55].

- **Thriving (7–10)**: Respondents in this category experience high life satisfaction, well-being, and optimism. Access to resources and support networks enhances their overall happiness and contentment [52–55].

### 2.7. Explanatory variables

The explanatory variables in the study were grouped into three distinct categories: individual, household, and community-level factors. Variables at the individual level included age, gender, educational attainment, demographic classification, income, and employment status; the number of persons living in the home, the number of individuals under the age of 18, child hunger, the SASSA social grant, the health facility, and medical aid were factors at the household level. Dwelling type, media accessibility, municipalities, and survey years are examples of community-level characteristics [28–34]. The availability of the variables and earlier research findings served as the basis for selecting these predictors [1,21]. Certain variables from the dataset retained their original categorization during this analysis, while others were meticulously redefined and recorded to enhance the study's precision [28–34]. Most selected variables were evaluated as straightforward binary measures, requiring "yes" or "no" responses. Additionally, some variables were derived by aggregating answers from multiple questions. Comprehensive descriptions of these variables are provided in another section [28–34] (Refer to Table 1 below).

### 2.8. How questions derived from the 2009–2024 GCRO QoL questionnaire instrument will be analyzed

Questions were retrieved from the Quality of Life I (2009–2020) and Quality of Life 7 (2023/2024) Survey: Full Questionnaire. The questions from the GCRO QoL instrument will be analyzed as presented in Table 2.

### 2.9. Data preparation and analysis

STATA version 2021 statistical software was used for data management and analysis. The study examined the individual-level, household-level and community-level factors determining life satisfaction among South African nationals aged 18 years or above, residing in the Gauteng Province. The dataset was analyzed according to the study's objectives and specific measures. Weighting variables from each survey were utilized alongside the "svy command" to address sampling imbalances and account for over- and under-representation. This approach also accommodated the intricate survey design and enhanced the generalizability of the results. The findings derived from the 2009–2024 datasets were presented as follows: First, univariate analysis was carried out to examine the distribution and characteristics using the descriptive statistics to report the demographic characteristics of the respondents, percentage and frequency distributions, and visualizations (vertical bar chart, stacked bar chart, grouped/clustered bar chart) (specific objective 1) (Figs 1–3, and Table 3). Second, bivariate analysis using Chi-square test of associations was employed to test the associations between life satisfaction and demographic determinants stratified by sex using 5% level of significance threshold (specific objective 2) (Table 4). Third, pathway analysis (or pathway model), mediation analysis and generalized structural equation modelling (gSEM) were employed to analyze the specific objective 3. Mediation analysis was utilized to explain

**Table 1. Detailed explanations of individual-level, household-level and community-level variables categorization.**

| Individual-level factors | Categorization |
|---|---|
| Age | 1 = 18–27; 2 = 28–37; 3 = 38–47; 4 = 48 + years |
| Sex | 1 = male; 2 = female |
| Highest education | 1 = no education; 2 = primary; 3 = secondary; 4 = higher |
| Population group | 1 = Black African; 2 = Coloured; 3 = White; 4. Indian/Asian |
| Respondent's income | 1 = no income; 2 = low (R1–R12,800); 3 = middle (R12,801–R25,600); 4 = high (R25,601+) |
| Employment | 1 = not working; 2 = working |
| Medical aid | 1 = no; 2 = yes |
| Healthcare facility | 1 = public; 2 = private; 3 = both public and private; 4 = traditional |
| Life satisfaction | 0 = suffering; 1 = struggling; 2 = thriving |
| **Household-level factors** | |
| Number of people living in H/H | 1 = 1; 2 = 2; 3 = 3; 4 = 4 + persons |
| Number of people under 18 in H/H | 1 = 0; 2 = 1; 3 = 2; 4 = 3; 5 = 4 + persons |
| Not enough to feed children | 1 = No; 2 = Yes; 3 = No children in H/H |
| Receiving SASSA social grant | 1 = No; 2 = Yes |
| **Community-level factor** | |
| Type of dwelling | 1 = Formal; 2 = Informal |
| Basic services | 1 = No; 2 = Yes |
| Municipality | 1 = City of Ekurhuleni; 2 = City of Johannesburg; 3 = City of Tshwane; 4 = Emfuleni; 5 = Lesedi; 6 = Midvaal; 7 = Merafong; 8 = Mogale City; 9 = Rand West |
| Survey years | 1 = 2009; 2 = 2011; 3 = 2013–2014; 4 = 2015–2016; 5 = 2017–2018; 6 = 2020–2021; 7 = 2023–2024 |

Source: Authors' compilation, 2024.

the effects of the explanatory variable on the outcome variable (Table 5), while the pathway model was used to identify the direct, and indirect effects of explanatory variables on the outcome variable (Figs 4 and 5), and the generalized structural equation modelling (gSEM) was used to study the effects of the determinants (education, employment, income, and survey years) on life satisfaction as the outcome variable; the variables were either ordinal or binary in nature (Tables 6 and 7) as well as the percentages of the mediation analysis (Table 8). Therefore, combining the pathway models with gSEM will create more robust and comprehensive models to study the direct and indirect effects influencing life satisfaction [61–63]. Further, nonlinear combinations of parameter estimates (nlcom) were generated using the nlcom command on STATA to generate direct and total effects between the outcome, mediator, and independent variables [64,65], and the gSEM and nlcom were both stratified by sex, with statistical significance of $\rho < 0.05$ signifying level of precision (specific objective 4).

## 2.10. Handling of missing data and bootstrapping in mediation analysis

Missing data in this study were carefully assessed to maintain methodological rigor. Irrelevant variables were excluded, and only relevant ones retained for analysis. The extent of missingness in key variables was examined using descriptive statistics, including frequencies and percentages. To address potential biases, bootstrapping with 5,000 resamples was applied in the Generalized Structural Equation Modeling (GSEM) and mediation analyses, improving the stability and reliability of indirect effect estimates. Chi-square tests confirmed that missing data were randomly distributed and not systematically associated with demographic variables. Visual tools such as charts was used to explore missing data patterns across respondent groups, and these procedures ensured that the mediation analysis remained robust, replicable, and free from distortions, strengthening the credibility of the findings.

**Table 2. Distribution of variables showing the instrument variable names and how they were analyzed.**

| S/N | Variable Description | Variable Name | Variable Analysis |
|-----|----------------------|---------------|-------------------|
| 1 | What is the sex of the respondent? | a2_sex | Categorization of variables and the use of descriptive statistics for analysis |
| 2 | How old are you? | q14_2_age | Categorization of variables and the use of summary and descriptive statistics for analysis |
| 3 | To which population group does the respondent belong? | a1_pop_group | Categorization of variables and the use of summary and descriptive statistics for analysis |
| 4 | What is the highest level of education you have completed? | q14_1_education | Categorization of variables and the use of summary and descriptive statistics for analysis |
| 5 | What is the total amount of money brought into the household per month by all household members? Please include all the money coming into the household, from all sources. This includes grants, help from friends and family, and rental or business income. Please exclude amounts deducted for tax, medical aid and pension contributions. | q15_3_income | Categorization of variables and the use of summary and descriptive statistics for analysis |
| 6 | In the past 7 days, did you do any type of work, business, or activity for which you got paid or expected to be paid (even if just for one hour)? This includes selling things, piece work, and work that's part of your own business. | q10_2_working | Categorisation of variables and the use of summary and descriptive statistics for analysis |
| 7 | Are you personally covered by any form of medical aid or other medical insurance? | q13_5_medical_aid | Categorisation of variables and the use of summary and descriptive statistics for analysis |
| 8 | Where do you usually go for health care? | q13_1_healthcare | Categorisation of variables and the use of summary and descriptive statistics for analysis |
| 9 | How many people, including you, live in this household? Please include babies and children. | q14_5_people | Categorisation of variables and the use of summary and descriptive statistics for analysis |
| 10 | How many members of this household are under 18 years old? | q14_6_under18 | Categorisation of variables and the use of summary and descriptive statistics for analysis |
| 11 | In the past 12 months, has there ever been a time when there was not enough money to feed the children in the household? | q6_5_feed_children | Categorisation of variables and the use of summary and descriptive statistics for analysis |
| 12 | Does anybody in this household receive a social grant of any type, such as an old age pension, child care or disability grant? | q14_10_social_grant | Categorisation of variables and the use of summary and descriptive statistics for analysis |
| 13 | Which type of dwelling does this household occupy? | a3_dwelling_type | Categorisation of variables and the use of summary and descriptive statistics for analysis |
| 14 | Does this household have any of the following that are in good working order, that is not broken? | q6_3_1_landline; q6_3_2_cell; q6_3_3_tv; q6_3_4_computer; q6_3_5_radio; q6_3_6_dstv; q6_3_7_internet; | Categorisation of variables and the use of summary and descriptive statistics for analysis |
| 15 | Region/Municipality Name | municipality_coded | Categorisation of variables and the use of summary and descriptive statistics for analysis |
| 16 | How satisfied are you with your life AS A WHOLE these days? | q9_9_life | Bivariate and multivariate regression analysis will be employed |
| 17 | In which province or country were you born? | q3_1_birth_prov | Will be used to filter for South African nationals |

**Source:** Authors' compilation, 2024.

## 2.11. Ethics

The Gauteng City-Region Observatory (GCRO) obtained ethical clearance from the Human Research Ethics Committee (Non-Medical) at the University of the Witwatersrand, Johannesburg (clearance number [H19/11/19]). Verbal consent was transparently obtained from participants aged 18 and above. All respondents were informed of the voluntary nature of participation and assured of confidentiality and anonymity throughout the study.

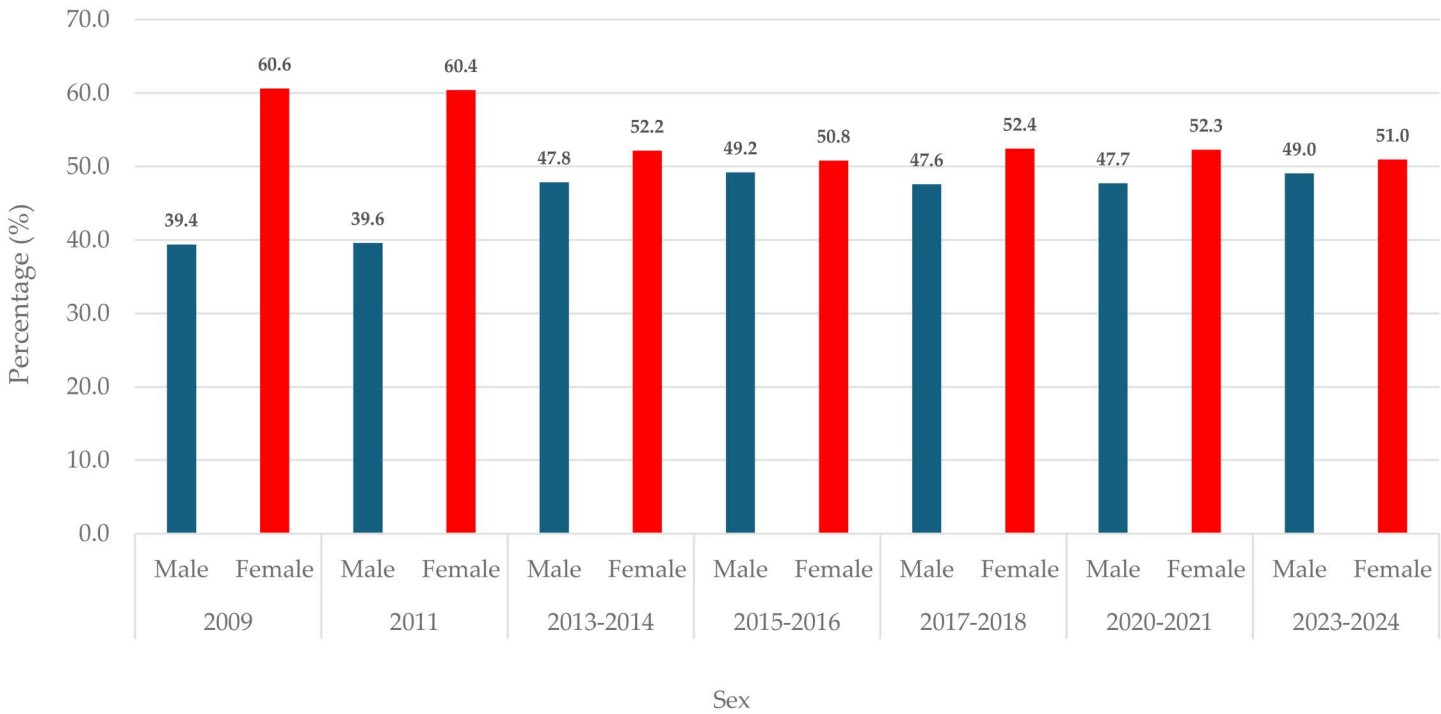

**Fig 1. Sex distribution of respondents.** The vertical bar chart depicts the percentage distribution of male and female respondents residing in Gauteng, South Africa. The chart illustrates the percentage distribution of observations for each data point or the grouping of the data points of sex by survey years (2009–2024).

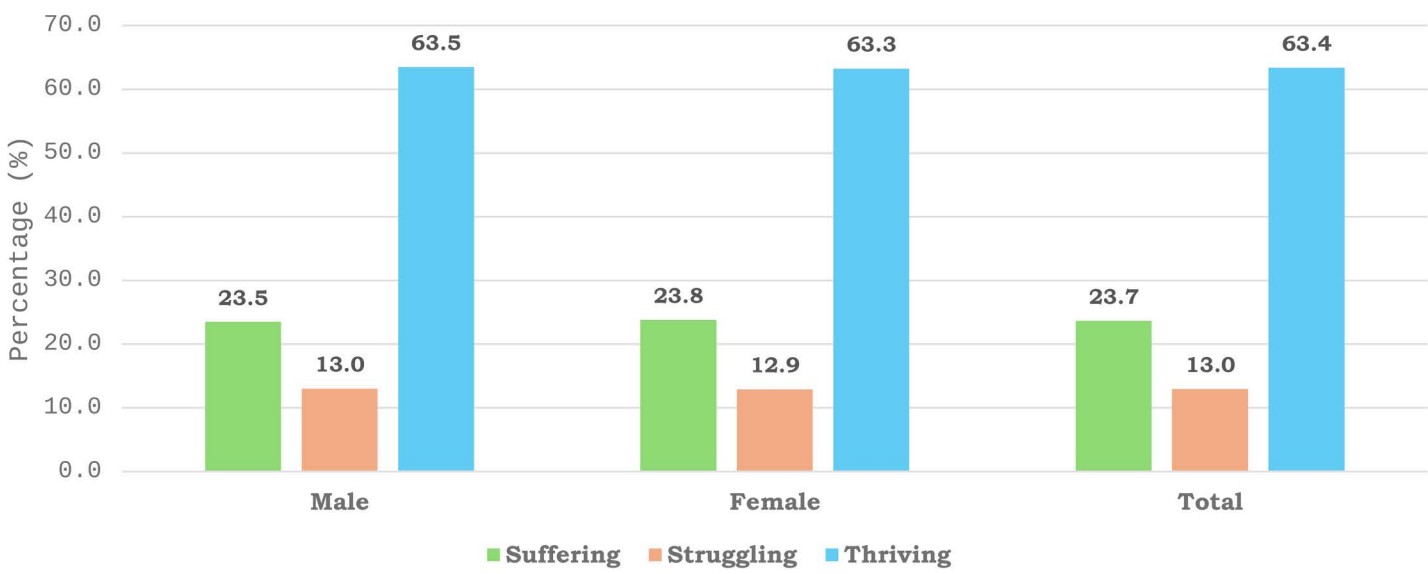

**Fig 2. Distribution of life satisfaction levels—suffering, struggling, and thriving—by sex.** The grouped bar chart shows that the majority of both male and female respondents reported thriving life satisfaction in Gauteng Province, South Africa. The percentages were calculated from a representative sample, with life satisfaction levels stratified by sex (male and female).

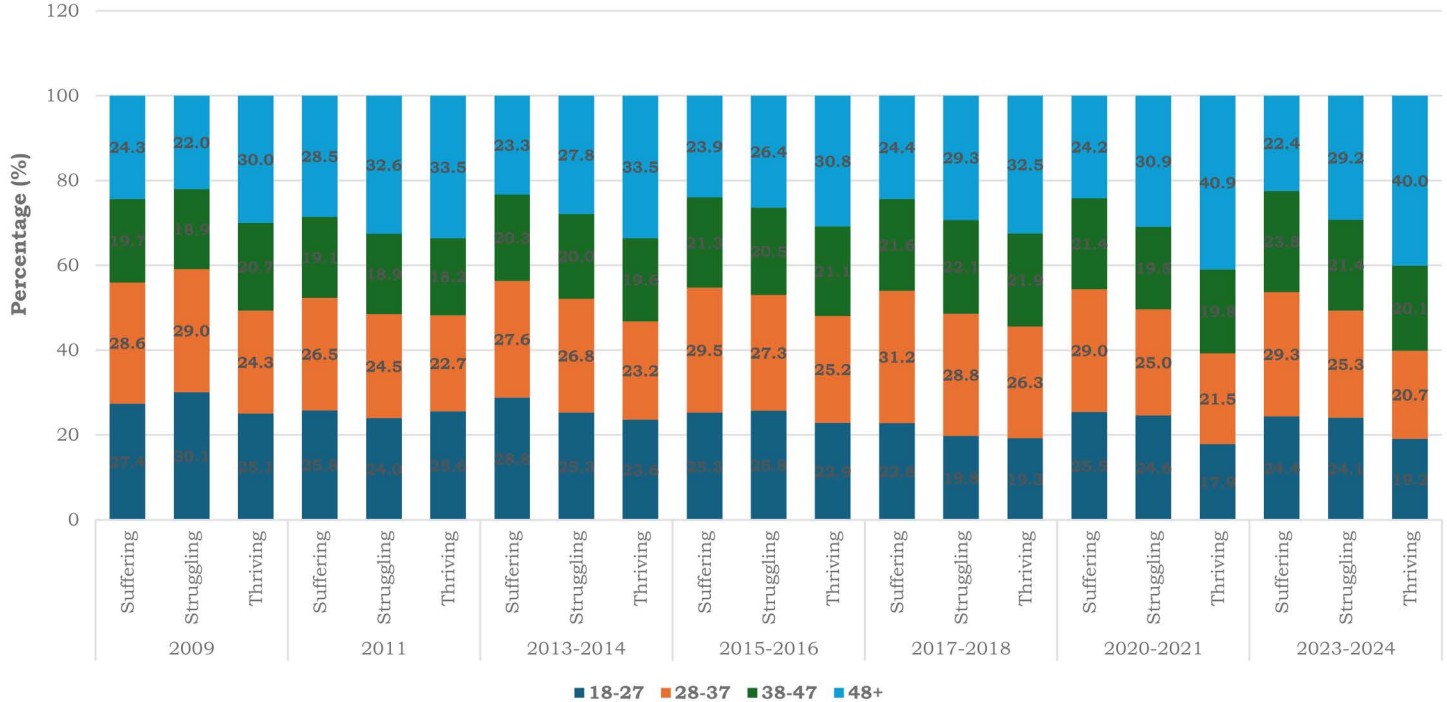

**Fig 3. Shows the life satisfaction levels by age as reported by the respondents who are South Africans, residing in Gauteng province, South Africa.** The percentage was determined from the representative sample of the number of life satisfaction scales distributed by the different ages across the survey years (2009–2024).

## 2.12. Ethics approval and consent to participate

The Human Research Ethics Committee (non-medical) at the University of the Witwatersrand, through the Johannesburg Research Office, granted ethical clearance to the GCRO organisation for the design, procedures, and survey instruments used in its regular Quality of Life (QoL) surveys. Additionally, the University of the Witwatersrand's Institutional Review Board (IRB) in Johannesburg reviewed the GCRO QoL survey methodology. Before taking part in the study, all human subjects provided their informed consent, according to the GCRO Field Staff. The Human Research Ethics Committee (non-medical), University of the Witwatersrand, Johannesburg, has examined and approved the protocols and questions for the standard GCRO QoL surveys. This study has no human volunteers; it only uses secondary data. Consequently, no official ethical approval was needed. However, to utilise the 2009–2021 GCRO QoL survey datasets from the GCRO repository (https://www.datafirst.etc.ac.za/dataportal/index.php/collections/GCRO), authorisation was secured by submitting a written request to the GCRO. The study was carried out in accordance with the Declaration of Helsinki and the applicable ethical standards and laws. The Declaration of Helsinki and all applicable ethical rules and regulations were followed in the conduct of the study. The University of the Witwatersrand, Johannesburg GCRO QoL surveys' Ethics Committee (Project identification code: Round 1 GCRO QoL 2009; Round 2 GCRO QoL 2011; Round 3 GCRO QoL 2013–2014; Round 4 GCRO QoL 2015–2016; Round 5 GCRO QoL 2017–2018; Round 6 GCRO QoL 2020–2021, and Round 6 GCRO QoL 2021–2024) approved the protocol. Only the use of the data for this research topic and the publication of the results in a peer-reviewed journal were conditions of the permission granted.

**Table 3. Socio-demographic Characteristics of Respondents stratified by Sex (n = 70,314).**

| Demographics | Male (n = 33,325) | | Female (n = 36,989) | | Total (N = 70,314) | |
|---|---|---|---|---|---|---|
| **Individual-level factors** | | | | | | |
| **Life satisfaction** | **Frequency** | **%** | **Frequency** | **%** | **Frequency** | **%** |
| Suffering | 7826 | 23.5 | 8813 | 23.8 | 16639 | 23.7 |
| Struggling | 4342 | 13.0 | 4779 | 12.9 | 9121 | 13.0 |
| Thriving | 21157 | 63.5 | 23397 | 63.3 | 44554 | 63.4 |
| **Age group** | | | | | | |
| 18-27 | 7806 | 23.4 | 8046 | 21.8 | 15852 | 22.5 |
| 28-37 | 8742 | 26.2 | 9422 | 25.5 | 18164 | 25.8 |
| 38-47 | 7022 | 21.1 | 7453 | 20.1 | 14475 | 20.6 |
| 48+ | 9755 | 29.3 | 12068 | 32.6 | 21824 | 31.0 |
| **Highest education** | | | | | | |
| No education | 557 | 1.7 | 880 | 2.4 | 1437 | 2.0 |
| Primary | 3223 | 9.7 | 4377 | 11.8 | 7600 | 10.8 |
| Secondary | 21826 | 65.5 | 24444 | 66.1 | 46270 | 65.8 |
| Higher | 7719 | 23.2 | 7288 | 19.7 | 15007 | 21.3 |
| **Pop group** | | | | | | |
| Black African | 27670 | 83.0 | 30692 | 83.0 | 58362 | 83.0 |
| Coloured | 1033 | 3.1 | 1256 | 3.4 | 2289 | 3.3 |
| Indian/Asian | 762 | 2.3 | 820 | 2.2 | 1581 | 2.2 |
| White | 3860 | 11.6 | 4221 | 11.4 | 8082 | 11.5 |
| **Income** | | | | | | |
| No income | 2203 | 6.6 | 1834 | 5.0 | 4037 | 5.7 |
| Low | 24762 | 74.3 | 29448 | 79.6 | 54210 | 77.1 |
| Middle | 3456 | 10.4 | 3257 | 8.8 | 6713 | 9.5 |
| High | 2904 | 8.7 | 2450 | 6.6 | 5354 | 7.6 |
| **Employment** | | | | | | |
| Not employed | 18893 | 56.7 | 26044 | 70.4 | 44937 | 63.9 |
| Employed | 14432 | 43.3 | 10945 | 29.6 | 25377 | 36.1 |
| **Household-level factors** | | | | | | |
| **HH members** | | | | | | |
| 1 | 8111 | 24.3 | 3557 | 9.6 | 11668 | 16.6 |
| 2 | 6308 | 18.9 | 6542 | 17.7 | 12850 | 18.3 |
| 3 | 5357 | 16.1 | 7078 | 19.1 | 12435 | 17.7 |
| 4 and above | 13549 | 40.7 | 19812 | 53.6 | 33361 | 47.4 |
| **Children under 18 in HH** | | | | | | |
| None | 17802 | 53.4 | 11846 | 32.0 | 29648 | 42.2 |
| 1 | 5746 | 17.2 | 8094 | 21.9 | 13840 | 19.7 |
| 2 | 5256 | 15.8 | 8466 | 22.9 | 13722 | 19.5 |
| 3 | 3224 | 9.7 | 5525 | 14.9 | 8749 | 12.4 |
| 4+ | 1297 | 3.9 | 3058 | 8.3 | 4355 | 6.2 |
| **Child hunger** | | | | | | |
| No | 19279 | 57.9 | 24312 | 65.7 | 43591 | 62.0 |
| Yes | 3811 | 11.4 | 6506 | 17.6 | 10317 | 14.7 |
| No children in HH | 10235 | 30.7 | 6171 | 16.7 | 16406 | 23.3 |
| **Social Grant** | | | | | | |
| No | 20187 | 60.6 | 15257 | 41.2 | 35444 | 50.4 |
| Yes | 13138 | 39.4 | 21732 | 58.8 | 34870 | 49.6 |

*(Continued)*

**Table 3.** (Continued)

| Demographics | Male (n = 33,325) | | Female (n = 36,989) | | Total (N = 70,314) | |
|---|---|---|---|---|---|---|
| **Health facility** | | | | | | |
| Public | 11429 | 34.3 | 11343 | 30.7 | 22772 | 32.4 |
| Private | 18236 | 54.7 | 22062 | 59.6 | 40298 | 57.3 |
| Both public and private | 3216 | 9.6 | 3369 | 9.1 | 6585 | 9.4 |
| Traditional | 444 | 1.3 | 215 | 0.6 | 659 | 0.9 |
| **Medical aid** | | | | | | |
| No | 27839 | 83.5 | 31683 | 85.7 | 59522 | 84.7 |
| Yes | 5486 | 16.5 | 5306 | 14.3 | 10792 | 15.3 |
| **Community-level factors** | | | | | | |
| **Dwelling type** | | | | | | |
| Formal | 27931 | 83.8 | 31510 | 85.2 | 59441 | 84.5 |
| Informal | 5394 | 16.2 | 5479 | 14.8 | 10873 | 15.5 |
| **Access to media** | | | | | | |
| No | 426 | 1.3 | 380 | 1.0 | 806 | 1.1 |
| Yes | 32899 | 98.7 | 36609 | 99.0 | 69508 | 98.9 |
| **Municipality** | | | | | | |
| City of Ekurhuleni | 8927 | 26.8 | 9692 | 26.2 | 18619 | 26.5 |
| City of Johannesburg | 11727 | 35.2 | 12870 | 34.8 | 24597 | 35.0 |
| City of Tshwane | 8024 | 24.1 | 8956 | 24.2 | 16980 | 24.1 |
| Emfuleni | 1958 | 5.9 | 2344 | 6.3 | 4302 | 6.1 |
| Lesedi | 271 | 0.8 | 324 | 0.9 | 595 | 0.8 |
| Midvaal | 264 | 0.8 | 245 | 0.7 | 509 | 0.7 |
| Merafong | 502 | 1.5 | 619 | 1.7 | 1121 | 1.6 |
| Mogale City | 990 | 3.0 | 1163 | 3.1 | 2153 | 3.1 |
| Rand West | 662 | 2.0 | 776 | 2.1 | 1438 | 2.0 |
| **Survey year** | | | | | | |
| 2009 | 1888 | 5.7 | 2907 | 7.9 | 4795 | 6.8 |
| 2011 | 906 | 2.7 | 1381 | 3.7 | 2287 | 3.3 |
| 2013-2014 | 8044 | 24.1 | 8769 | 23.7 | 16813 | 23.9 |
| 2015-2016 | 8136 | 24.4 | 8401 | 22.7 | 16537 | 23.5 |
| 2017-2018 | 6211 | 18.6 | 6846 | 18.5 | 13057 | 18.6 |
| 2020-2021 | 4036 | 12.1 | 4421 | 12.0 | 8457 | 12.0 |
| 2023-2024 | 4104 | 12.3 | 4264 | 11.5 | 8368 | 11.9 |
| **Total** | 33325 | 100 | 36989 | 100 | 70314 | 100 |

### 2.13. Institutional review board statement

The GCRO obtains ethical clearance from the Human Research Ethics Committee (non-medical) from the University of the Witwatersrand, Johannesburg Research Office. Verbal consent was obtained from the respondents aged 18 +. All respondents were informed about the voluntary nature of participation, including confidentiality and anonymity.

## 3. Results

Table 3 presents the weighted sample of 70,314 respondents stratified by sex (male: 33,325 [47.4%]; female: 36,989 [52.6%]) across survey years (2009–2024) in Gauteng province, South Africa. It also summarizes respondents' demographic characteristics, organized into three categories: individual-level, household-level, and community-level factors.

**Table 4. Chi-square analysis showing the associations between life satisfaction and factor-related indicators (individual, household, and community) among male and female respondents (n = 70,314).**

| | Males | | | | | | | | Females | | | | | | | |
|---|---|---|---|---|---|---|---|---|---|---|---|---|---|---|---|---|
| Individual-level factors | Suffering | | Struggling | | Thriving | | X² | p-value | Suffering | | Struggling | | Thriving | | X² | p-value |
| Age group | Fre-quency | % | Fre-quency | % | Fre-quency | % | | | Fre-quency | % | Fre-quency | % | Fre-quency | % | | |
| 18-27 | 2,020 | 25.9 | 1,096 | 14.0 | 4,690 | 60.1 | 248.519 | 0.000 | 2,170 | 27.0 | 1,098 | 13.6 | 4,777 | 59.4 | 412.362 | 0.000 |
| 28-37 | 2,364 | 27.0 | 1,228 | 14.0 | 5,150 | 58.9 | | | 2,578 | 27.4 | 1,282 | 13.6 | 5,561 | 59.0 | | |
| 38-47 | 1,651 | 23.5 | 945 | 13.5 | 4,427 | 63.0 | | | 1,862 | 25.0 | 945 | 12.7 | 4,646 | 62.3 | | |
| 48+ | 1,791 | 18.4 | 1,073 | 11.0 | 6,891 | 70.6 | | | 2,203 | 18.3 | 1,454 | 12.0 | 8,412 | 69.7 | | |
| **Highest education** | | | | | | | | | | | | | | | | |
| No education | 166 | 29.9 | 62 | 11.1 | 329 | 59.1 | 637.891 | 0.000 | 270 | 30.6 | 113 | 12.8 | 498 | 56.5 | 445.155 | 0.000 |
| Primary | 871 | 27.0 | 416 | 12.9 | 1,937 | 60.1 | | | 1,070 | 24.4 | 570 | 13.0 | 2,738 | 62.5 | | |
| Secondary | 5,743 | 26.3 | 3,009 | 13.8 | 13,074 | 59.9 | | | 6,392 | 26.1 | 3,239 | 13.3 | 14,813 | 60.6 | | |
| Higher | 1,046 | 13.6 | 855 | 11.1 | 5,817 | 75.4 | | | 1,082 | 14.8 | 858 | 11.8 | 5,349 | 73.4 | | |
| **Population group** | | | | | | | | | | | | | | | | |
| Black African | 7,298 | 26.4 | 3,791 | 13.7 | 16,580 | 59.9 | 890.601 | 0.000 | 8,232 | 26.8 | 4,177 | 13.6 | 18,284 | 59.6 | 1,000.000 | 0.000 |
| Coloured | 192 | 18.6 | 128 | 12.4 | 713 | 69.0 | | | 246 | 19.6 | 176 | 14.0 | 833 | 66.4 | | |
| Indian/Asian | 92 | 12.1 | 95 | 12.5 | 574 | 75.4 | | | 94 | 11.5 | 80 | 9.8 | 645 | 78.7 | | |
| White | 244 | 6.3 | 327 | 8.5 | 3,289 | 85.2 | | | 240 | 5.7 | 347 | 8.2 | 3,634 | 86.1 | | |
| **Income** | | | | | | | | | | | | | | | | |
| No income | 807 | 36.6 | 274 | 12.5 | 1,122 | 50.9 | 1,200.000 | 0.000 | 579 | 31.6 | 240 | 13.1 | 1,015 | 55.4 | 1,100.000 | 0.000 |
| Low | 6,449 | 26.0 | 3,451 | 13.9 | 14,862 | 60.0 | | | 7,744 | 26.3 | 4,011 | 13.6 | 17,693 | 60.1 | | |
| Middle | 368 | 10.6 | 395 | 11.4 | 2,693 | 77.9 | | | 342 | 10.5 | 331 | 10.2 | 2,583 | 79.3 | | |
| High | 203 | 7.0 | 221 | 7.6 | 2,480 | 85.4 | | | 147 | 6.0 | 197 | 8.1 | 2,105 | 85.9 | | |
| **Employment** | | | | | | | | | | | | | | | | |
| Not employed | 4,905 | 26.0 | 2,444 | 12.9 | 11,543 | 61.1 | 136.461 | 0.000 | 6,558 | 25.2 | 3,304 | 12.7 | 16,182 | 62.1 | 95.314 | 0.000 |
| Employed | 2,921 | 20.2 | 1,898 | 13.2 | 9,613 | 66.6 | | | 2,255 | 20.6 | 1,475 | 13.5 | 7,215 | 65.9 | | |
| **Household-level factors** | | | | | | | | | | | | | | | | |
| **HH members** | | | | | | | | | | | | | | | | |
| 1 | 2,117 | 26.1 | 1,099 | 13.5 | 4,895 | 60.4 | 70.421 | 0.000 | 757 | 21.3 | 487 | 13.7 | 2,313 | 65.0 | 98.794 | 0.000 |
| 2 | 1,365 | 21.6 | 785 | 12.4 | 4,158 | 65.9 | | | 1,289 | 19.7 | 807 | 12.3 | 4,446 | 68.0 | | |
| 3 | 1,119 | 20.9 | 677 | 12.6 | 3,560 | 66.5 | | | 1,599 | 22.6 | 925 | 13.1 | 4,554 | 64.3 | | |
| 4 and above | 3,225 | 23.8 | 1,781 | 13.1 | 8,544 | 63.1 | | | 5,168 | 26.1 | 2,561 | 12.9 | 12,084 | 61.0 | | |
| **Children under 18 in HH** | | | | | | | | | | | | | | | | |
| None | 4,135 | 23.2 | 2,305 | 12.9 | 11,361 | 63.8 | 48.661 | 0.000 | 2,349 | 19.8 | 1,480 | 12.5 | 8,018 | 67.7 | 252.620 | 0.000 |
| 1 | 1,299 | 22.6 | 755 | 13.1 | 3,693 | 64.3 | | | 1,855 | 22.9 | 1,065 | 13.2 | 5,174 | 63.9 | | |
| 2 | 1,186 | 22.6 | 665 | 12.7 | 3,405 | 64.8 | | | 2,110 | 24.9 | 1,091 | 12.9 | 5,264 | 62.2 | | |
| 3 | 815 | 25.3 | 453 | 14.0 | 1,956 | 60.7 | | | 1,499 | 27.1 | 777 | 14.1 | 3,249 | 58.8 | | |
| 4+ | 392 | 30.2 | 164 | 12.6 | 742 | 57.2 | | | 1,001 | 32.7 | 366 | 12.0 | 1,691 | 55.3 | | |

*(Continued)*

| Individual-level factors | Males | | | | | | | | Females | | | | | | | |
|---|---|---|---|---|---|---|---|---|---|---|---|---|---|---|---|---|
| | Suffering | | Struggling | | Thriving | | X² | p-value | Suffering | | Struggling | | Thriving | | X² | p-value |
| Age group | Fre-quency | % | Fre-quency | % | Fre-quency | % | | | Fre-quency | % | Fre-quency | % | Fre-quency | % | | |
| **Child hunger** | | | | | | | | | | | | | | | | |
| No | 3,848 | 20.0 | 2,603 | 13.5 | 12,828 | 66.5 | 698.616 | 0.000 | 5,058 | 20.8 | 3,266 | 13.4 | 15,988 | 65.8 | 1,400.000 | 0.000 |
| Yes | 1,563 | 41.0 | 496 | 13.0 | 1,752 | 46.0 | | | 2,661 | 40.9 | 844 | 13.0 | 3,001 | 46.1 | | |
| No children in HH | 2,416 | 23.6 | 1,242 | 12.1 | 6,577 | 64.3 | | | 1,094 | 17.7 | 670 | 10.8 | 4,407 | 71.4 | | |
| **Social Grant** | | | | | | | | | | | | | | | | |
| No | 4,405 | 21.8 | 2,529 | 12.5 | 13,252 | 65.6 | 88.228 | 0.000 | 2,818 | 18.5 | 1,868 | 12.2 | 10,571 | 69.3 | 375.864 | 0.000 |
| Yes | 3,421 | 26.0 | 1,812 | 13.8 | 7,905 | 60.2 | | | 5,995 | 27.6 | 2,912 | 13.4 | 12,825 | 59.0 | | |
| **Health facility** | | | | | | | | | | | | | | | | |
| Public | 1,997 | 17.5 | 1,309 | 11.5 | 8,123 | 71.1 | 901.520 | 0.000 | 1,859 | 16.4 | 1,386 | 12.2 | 8,097 | 71.4 | 906.394 | 0.000 |
| Private | 5,342 | 29.3 | 2,497 | 13.7 | 10,397 | 57.0 | | | 6,470 | 29.3 | 2,858 | 13.0 | 12,734 | 57.7 | | |
| Both public and private | 368 | 11.4 | 441 | 13.7 | 2,406 | 74.8 | | | 442 | 13.1 | 492 | 14.6 | 2,434 | 72.3 | | |
| Traditional | 119 | 26.9 | 94 | 21.2 | 230 | 51.9 | | | 41 | 18.9 | 43 | 20.1 | 131 | 60.9 | | |
| **Medical aid** | | | | | | | | | | | | | | | | |
| No | 7,230 | 26.0 | 3,805 | 13.7 | 16,804 | 60.4 | 721.611 | 0.000 | 8,263 | 26.1 | 4,259 | 13.4 | 19,160 | 60.5 | 749.110 | 0.000 |
| Yes | 596 | 10.9 | 537 | 9.8 | 4,352 | 79.3 | | | 549 | 10.4 | 520 | 9.8 | 4,236 | 79.8 | | |
| **Community-level factors** | | | | | | | | | | | | | | | | |
| **Dwelling type** | | | | | | | | | | | | | | | | |
| Formal | 5,674 | 20.3 | 3,569 | 12.8 | 18,687 | 66.9 | 1,000.000 | 0.000 | 6,563 | 20.8 | 4,016 | 12.7 | 20,931 | 66.4 | 1,400.000 | 0.000 |
| Informal | 2,152 | 39.9 | 772 | 14.3 | 2,469 | 45.8 | | | 2,250 | 41.1 | 764 | 13.9 | 2,466 | 45.0 | | |
| **Access to media** | | | | | | | | | | | | | | | | |
| No | 182 | 42.7 | 52 | 12.2 | 192 | 45.0 | 117.279 | 0.000 | 161 | 42.3 | 54 | 14.1 | 166 | 43.6 | 88.412 | 0.000 |
| Yes | 7,644 | 23.2 | 4,289 | 13.0 | 20,965 | 63.7 | | | 8,652 | 23.6 | 4,726 | 12.9 | 23,231 | 63.5 | | |
| **Municipality** | | | | | | | | | | | | | | | | |
| City of Ekurhuleni | 2,308 | 25.9 | 1,153 | 12.9 | 5,465 | 61.2 | 153.315 | 0.000 | 2,573 | 26.5 | 1,173 | 12.1 | 5,946 | 61.3 | 218.616 | 0.000 |
| City of Johannes-burg | 2,551 | 21.8 | 1,643 | 14.0 | 7,533 | 64.2 | | | 2,909 | 22.6 | 1,842 | 14.3 | 8,120 | 63.1 | | |
| City of Tshwane | 1,815 | 22.6 | 923 | 11.5 | 5,286 | 65.9 | | | 1,965 | 21.9 | 1,064 | 11.9 | 5,926 | 66.2 | | |
| Emfuleni | 468 | 23.9 | 255 | 13.0 | 1,236 | 63.1 | | | 565 | 24.1 | 319 | 13.6 | 1,461 | 62.3 | | |
| Lesedi | 49 | 18.0 | 29 | 10.8 | 193 | 71.2 | | | 63 | 19.3 | 41 | 12.6 | 220 | 68.1 | | |
| Midvaal | 57 | 21.8 | 39 | 14.8 | 167 | 63.3 | | | 50 | 20.5 | 23 | 9.2 | 172 | 70.3 | | |
| Merafong | 133 | 26.5 | 78 | 15.5 | 291 | 58.0 | | | 182 | 29.4 | 78 | 12.7 | 359 | 58.0 | | |
| Mogale City | 257 | 26.0 | 147 | 14.9 | 586 | 59.2 | | | 289 | 24.9 | 155 | 13.3 | 718 | 61.8 | | |
| Rand West | 188 | 28.4 | 75 | 11.3 | 399 | 60.3 | | | 216 | 27.9 | 85 | 11.0 | 475 | 61.2 | | |

*(Continued)*

**Table 4.** (Continued)

| Individual-level factors | Males | | | | | | | | | | | | | | Females | | | | | | | | | | | | | |
|---|---|---|---|---|---|---|---|---|---|---|---|---|---|---|---|---|---|---|---|---|---|---|---|---|---|---|---|---|
| | Suffering | | Struggling | | Thriving | | X² | | p-value | | Suffering | | Struggling | | Thriving | | X² | | p-value |
| Age group | Fre-quency | % | Fre-quency | % | Fre-quency | % | | | | | Fre-quency | % | Fre-quency | % | Fre-quency | % | | |
| Survey year | | | | | | | | | | | | | | | | | | |
| 2009 | 575 | 30.4 | 405 | 21.4 | 909 | 48.1 | 543.243 | 0.000 | | | 1,105 | 38.0 | 531 | 18.3 | 1,271 | 43.7 | 984.975 | 0.000 |
| 2011 | 333 | 36.8 | 144 | 15.9 | 429 | 47.4 | | | | | 512 | 37.0 | 213 | 15.4 | 657 | 47.6 | | |
| 2013-2014 | 1,835 | 22.8 | 794 | 9.9 | 5,416 | 67.3 | | | | | 1,926 | 22.0 | 815 | 9.3 | 6,028 | 68.7 | | |
| 2015-2016 | 1,602 | 19.7 | 1,208 | 14.8 | 5,326 | 65.5 | | | | | 1,503 | 17.9 | 1,263 | 15.0 | 5,635 | 67.1 | | |
| 2017-2018 | 1,407 | 22.7 | 713 | 11.5 | 4,091 | 65.9 | | | | | 1,530 | 22.4 | 843 | 12.3 | 4,474 | 65.3 | | |
| 2020-2021 | 1,085 | 26.9 | 552 | 13.7 | 2,399 | 59.4 | | | | | 1,163 | 26.3 | 558 | 12.6 | 2,700 | 61.1 | | |
| 2023-2024 | 990 | 24.1 | 527 | 12.8 | 2,587 | 63.0 | | | | | 1,074 | 25.2 | 557 | 13.1 | 2,633 | 61.7 | | |

$X^2$ = Chi-square; Note ρ-value (ρ): *ρ < 0 001.

### 3.1. Percentage distribution of male and female respondents

The multiple bar chart shows the percentage distribution of respondents by sex across the survey years as shown in Fig 1. Across the survey years, the majority of the respondents who participated in the survey from 2009–2024 were female (Fig 1).

### 3.2. Percentage distribution of life satisfaction by sex

Fig 2 illustrates the distribution of life satisfaction scales, categorized by sex, among individuals residing in Gauteng province, South Africa. The grouped bar chart revealed slight differences in respondents' reported life satisfaction between males (63.5%) and females (63.3%) (Fig 2). This finding suggests that the slight difference in life satisfaction between males and females in Gauteng may be influenced by social, economic, and cultural factors, yet the minimal gap indicates overall comparable well-being across sexes.

### 3.3. Percentage distribution of respondents' age by life satisfaction

Fig 3 shows the proportion by age indicating the life satisfaction scale among South Africans residing in Gauteng province across the survey years. From the stacked bar graph, struggling and suffering life satisfaction scales were mostly reported among age groups 18–27 years and 38–47 years. However, thriving life satisfaction level were majorly reported by age groups 28–37 years and 48 + years (Fig 3).

### 3.4. Demographic characteristics of the respondents

For the individual-level, a majority of both male (29.3%) and female (32.6%) respondents were aged 48 years above, and more respondents were found to be educated to the secondary level (males—65.5% and females—66.1%). 83% of both male and female respondents were Black African (Table 3). Moreover, more than half of the respondents reported earning low income (male—74.3%; female – 79.6%) and were not employed (males—56.7%; females—70.4%). By household factors, male (40.7%) and female (53.6%) respondents have 4 and more household members, and more reported that they have no children under 18 (male—53.4%; female—32%). Also, a majority of the respondents reported having no medical aid (males—83.5%; females—85.7%) and attend a private health facility (males—54.7%; females—59.6%). Furthermore, when considering community-level factors, most respondents indicated that they lived in a formal dwelling place (males—83.8%; females—85.2%), have access to media (males—98.7%; females—99.0%), and are resident in major

**Table 5. Showing the gSEM of the determinants of life satisfaction levels among male respondents (n = 33,325).**

**Males**

| Life Satisfaction (Outcome) | Coef. (base outcome) | Robust Std Err. | z | P>|Z| | [95% Conf. Interval] | |
|---|---|---|---|---|---|---|
| **Struggling** | | | | | | |
| No education | **Ref.** | | | | | |
| Primary | 0.251 | 0.183 | 1.370 | 0.170 | −0.107 | 0.609 |
| Secondary | 0.304 | 0.171 | 1.780 | 0.075 | −0.031 | 0.638 |
| Higher | 0.553 | 0.181 | 3.060 | 0.002 | 0.199 | 0.908 |
| Survey years | −0.037 | 0.017 | −2.170 | 0.030 | −0.070 | −0.004 |
| Not employed | **Ref.** | | | | | |
| Employed | 0.140 | 0.050 | 2.780 | 0.005 | 0.041 | 0.239 |
| No income | **Ref.** | | | | | |
| Low | 0.436 | 0.086 | 5.070 | 0.000 | 0.267 | 0.604 |
| Middle | 1.009 | 0.125 | 8.050 | 0.000 | 0.763 | 1.255 |
| Higher | 0.953 | 0.150 | 6.350 | 0.000 | 0.659 | 1.247 |
| _cons | −1.419 | 0.206 | −6.890 | 0.000 | −1.823 | −1.015 |
| **Thriving** | | | | | | |
| No education | **Ref.** | | | | | |
| Primary | 0.124 | 0.123 | 1.000 | 0.316 | −0.118 | 0.365 |
| Secondary | 0.060 | 0.115 | 0.530 | 0.599 | −0.165 | 0.285 |
| Higher | 0.493 | 0.122 | 4.040 | 0.000 | 0.254 | 0.732 |
| Survey years | −0.007 | 0.011 | −0.610 | 0.543 | −0.029 | 0.015 |
| Not employed | **Ref.** | | | | | |
| Employment | 0.075 | 0.035 | 2.140 | 0.033 | 0.006 | 0.143 |
| No income | **Ref.** | | | | | |
| Low | 0.480 | 0.058 | 8.300 | 0.000 | 0.367 | 0.594 |
| Middle | 1.462 | 0.089 | 16.460 | 0.000 | 1.288 | 1.637 |
| Higher | 1.875 | 0.110 | 16.970 | 0.000 | 1.658 | 2.091 |
| _cons | 0.159 | 0.136 | 1.170 | 0.242 | −0.108 | 0.425 |
| **Survey years** | | | | | | |
| No education | **Ref.** | | | | | |
| Primary | −0.062 | 0.092 | −0.670 | 0.500 | −0.243 | 0.118 |
| Secondary | 0.104 | 0.086 | 1.220 | 0.224 | −0.064 | 0.273 |
| Higher | 0.138 | 0.089 | 1.560 | 0.120 | −0.036 | 0.313 |
| Not employed | **Ref.** | | | | | |
| Employment | −0.029 | 0.025 | −1.170 | 0.243 | −0.078 | 0.020 |
| No income | **Ref.** | | | | | |
| Low | 0.783 | 0.033 | 23.660 | 0.000 | 0.718 | 0.847 |
| Middle | 0.945 | 0.048 | 19.610 | 0.000 | 0.851 | 1.040 |
| Higher | 1.028 | 0.054 | 19.170 | 0.000 | 0.923 | 1.133 |
| _cons | 3.510 | 0.093 | 37.810 | 0.000 | 3.328 | 3.692 |
| var(e.year) | 2.476 | 0.021 | | | 2.435 | 2.517 |

Note: * p<0 05, ** p<0 01, *** p<0 001.

**Fig 4. Structural equation model for demographic factors (education, employment, and income), and life satisfaction (struggling and thriving) by sex (males) across the survey years.** * $p < 0.005$; ** $p \leq 0.001$.

municipalities of Gauteng—City of Ekurhuleni: male (26.8%); female (26.2%); City of Johannesburg: male (35.2%) and female (34.2%); and City of Tshwane: male (24.1%) and female (24.2%) (Table 3).

### 3.5. Respondent's life satisfaction and associated factors (Individual-, Household-, and Community-Levels)

Table 4 presents the association between life satisfaction and factors-related indicators (individual-level, household-level, and community-level), stratified by sex using Chi-square. The results showed that there is a significant association between life satisfaction levels and individual-level factors (age group, education, population group, income, and employment) ($p < 0.001$) (Table 4). Regarding household factors, the Chi-square findings revealed that there is a significant association between life satisfaction levels and household-level factors (HH members, children under 18 in HH, child hunger, social grant, health facility, and medical aid) ($p < 0.001$) (Table 4). Further, concerning community-level factors, the Chi-square results showed that there is a significant association between life satisfaction levels and community-level factors (dwelling type, access to media, municipality, and survey year) ($p < 0.001$) (Table 4).

### 3.6. Generalized structural equation modelling (gSEM)

The Generalized Structural Equation Modelling (gSEM) presented the estimates of the model parameters to infer the underlying relationships of the direct, indirect, and total effects to understand the relationships between the explanatory variables and life satisfaction. Tables 5 and 6 present the analysis of independent variables (with reference categories) and life satisfaction levels using gSEM, which comprehensively accounts for both direct and indirect effects.

   3.6.1. **GSEM showing the determinants and life satisfaction among male respondents.** Table 5 shows the analysis of the determinants (explanatory variables) and life satisfaction levels. As shown in Table 5, the findings revealed

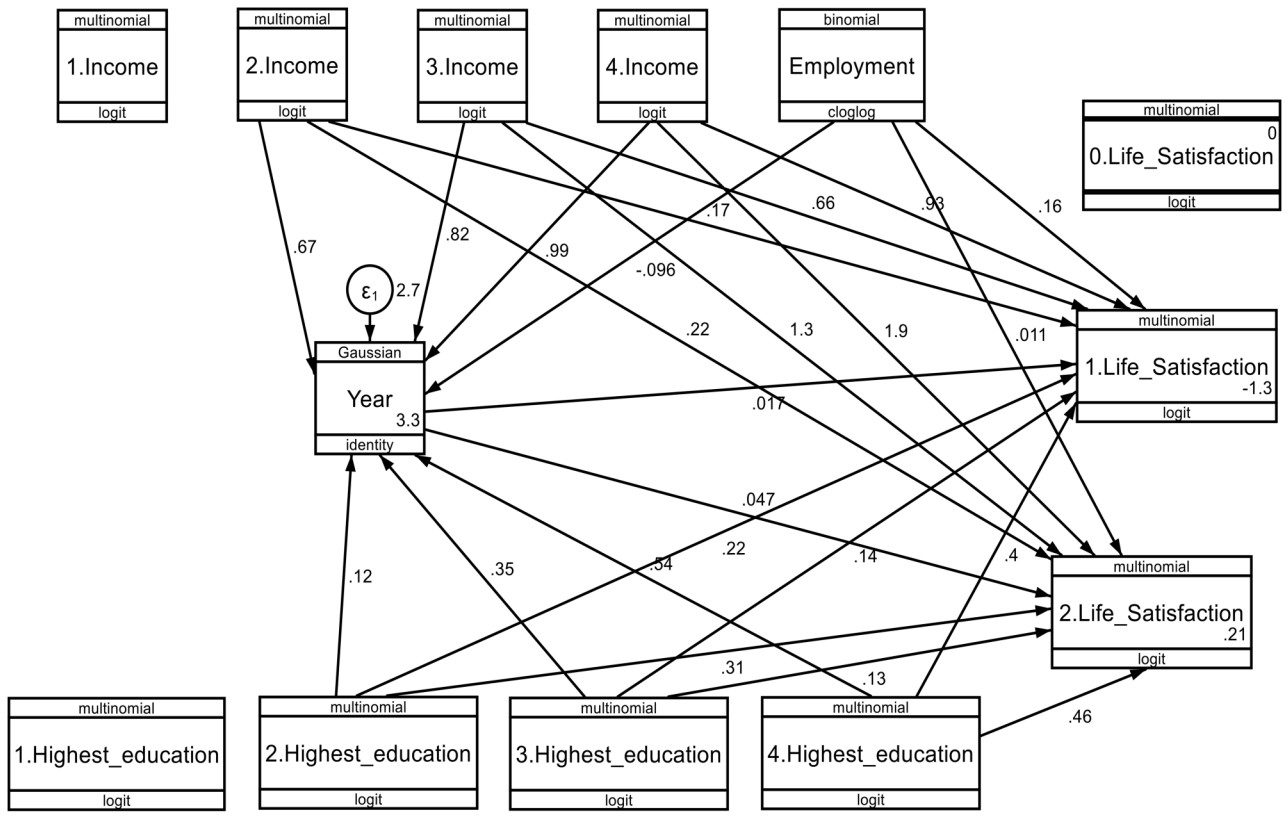

**Fig 5. Structural equation model for demographic factors (education, employment, and income), and life satisfaction (struggling and thriving) by sex (female) across the survey years.** * p < 0.05; ** p ≤ 0.01.

that demographic factors (education- higher: β = 0.553, SE = 0.181, 95% CI [0.199–0.908], ρ < 0.002), employment-working: β = −0.037, SE = 0.017, 95% CI [−0.070–0.004], ρ < 0.030), and income- low: β = 0.436, SE = 0.086, 95% CI [0.267–0.604], ρ < 0.000; middle: β = 1.009, SE = 0.125, 95% CI [0.763–1.255], ρ < 0.000); higher: β = 0.953, SE = 0.150, 95% CI [0.659–1.247], ρ < 0.000) significantly predicted struggling life satisfaction levels (Table 5). Also for thriving life satisfaction, demographic factors such as education (higher: β = 0.493, SE = 0.122, 95% CI [0.254–0.732], ρ < 0.000), employment (working: β = 0.075, SE = 0.035, 95% CI [0.006–0.143], ρ < 0.033), income (low: β = 0.480, SE = 0.058, 95% CI [0.367–0.594], ρ < 0.000; middle: β = 1.462, SE = 0.089, 95% CI [1.288–1.637], ρ < 0.000; higher: β = 1.875, SE = 0.110, 95% CI [1.658–2.091], ρ < 0.000) significantly predicted thriving life satisfaction levels (Table 5).

**3.6.2. GSEM showing the determinants and life satisfaction among female respondents.** Table 6 shows the analysis of the determinants (explanatory variables) and life satisfaction levels. As shown in Table 6, the findings revealed that demographic factors (education- higher: β = 0.396, SE = 0.129, 95% CI [0.144–0.648], p < 0.002), employment-employed: β = −0.156, SE = 0.045, 95% CI [0.067–0.245], p < 0.001), and income- low: β = 0.175, SE = 0.081, 95% CI [0.016–0.334], p < 0.031; middle: β = 0.664, SE = 0.123, 95% CI [0.422–0.906], p < 0.000); higher: β = 0.932, SE = 0.155, 95% CI [0.628–1.236], p < 0.000) predicted struggling life satisfaction levels (Table 6). Also, for thriving life satisfaction, the findings showed that demographic factors such as education (primary: β = 0.309, SE = 0.086, 95% CI [0.140–0.477], ρ < 0.000; higher: β = 0.456, SE = 0.088, 95% CI [0.284–0.629], p < 0.000), and income (low: β = 0.221, SE = 0.057, 95% CI [0.109–0.332], p < 0.000; middle: β = 1.295, SE = 0.089, 95% CI [1.121–1.469], p < 0.000; high: β = 1.872, SE = 0.118, 95% CI [1.640–2.103], p < 0.000) significantly predicted thriving life satisfaction levels (Table 6).

**Table 6. Showing the gSEM of the determinants of life satisfaction levels among female respondents (n = 36,989).**

| Life Satisfaction (Outcome) | Coef. (base outcome) | Robust Std. Err. | z | p > \|Z\| | [95% Conf. Interval] | |
|---|---|---|---|---|---|---|
| **Females** | | | | | | |
| **Struggling** | | | | | | |
| No education | **Ref.** | | | | | |
| Primary | 0.224 | 0.127 | 1.760 | 0.078 | −0.025 | 0.472 |
| Secondary | 0.139 | 0.117 | 1.180 | 0.238 | −0.091 | 0.368 |
| Higher | 0.396 | 0.129 | 3.080 | 0.002 | 0.144 | 0.648 |
| Survey years | 0.017 | 0.014 | 1.210 | 0.228 | −0.010 | 0.044 |
| Not employed | **Ref.** | | | | | |
| Employed | 0.156 | 0.045 | 3.440 | 0.001 | 0.067 | 0.245 |
| No income | **Ref.** | | | | | |
| Low | 0.175 | 0.081 | 2.160 | 0.031 | 0.016 | 0.334 |
| Middle | 0.664 | 0.123 | 5.380 | 0.000 | 0.422 | 0.906 |
| Higher | 0.932 | 0.155 | 6.010 | 0.000 | 0.628 | 1.236 |
| _cons | −1.275 | 0.152 | −8.410 | 0.000 | −1.572 | −0.978 |
| **Thriving** | | | | | | |
| No education | **Ref.** | | | | | |
| Primary | 0.309 | 0.086 | 3.600 | 0.000 | 0.140 | 0.477 |
| Secondary | 0.131 | 0.079 | 1.660 | 0.096 | −0.023 | 0.285 |
| Higher | 0.456 | 0.088 | 5.180 | 0.000 | 0.284 | 0.629 |
| Survey years | 0.047 | 0.010 | 4.740 | 0.000 | 0.028 | 0.066 |
| Not employed | **Ref.** | | | | | |
| Employment | 0.011 | 0.033 | 0.340 | 0.733 | −0.054 | 0.077 |
| No income | **Ref.** | | | | | |
| Low | 0.221 | 0.057 | 3.880 | 0.000 | 0.109 | 0.332 |
| Middle | 1.295 | 0.089 | 14.590 | 0.000 | 1.121 | 1.469 |
| Higher | 1.872 | 0.118 | 15.850 | 0.000 | 1.640 | 2.103 |
| _cons | 0.205 | 0.105 | 1.950 | 0.052 | −0.001 | 0.412 |
| **Survey years** | | | | | | |
| No education | **Ref.** | | | | | |
| Primary | 0.118 | 0.068 | 1.730 | 0.083 | −0.015 | 0.251 |
| Secondary | 0.354 | 0.062 | 5.670 | 0.000 | 0.231 | 0.476 |
| Higher | 0.537 | 0.067 | 8.030 | 0.000 | 0.406 | 0.668 |
| Not employed | **Ref.** | | | | | |
| Employment | −0.096 | 0.025 | −3.860 | 0.000 | −0.145 | −0.047 |
| No income | **Ref.** | | | | | |
| Low | 0.672 | 0.034 | 20.040 | 0.000 | 0.606 | 0.737 |
| Middle | 0.821 | 0.052 | 15.870 | 0.000 | 0.719 | 0.922 |
| Higher | 0.985 | 0.056 | 17.560 | 0.000 | 0.875 | 1.095 |
| _cons | 3.321 | 0.072 | 45.940 | 0.000 | 3.179 | 3.463 |
| var(e.year) | 2.692 | 0.020 | | | 2.653 | 2.731 |

**Note: * p<0 05, ** p<0 01, *** p<0 001.**

### 3.7. Pathway model (pathway analysis) of the structural equation modelling (SEM)

The pathway model of the structural equation modelling (SEM) showed the impact of selected determinants (education, employment, income) on life satisfaction, with survey years, to estimate the direct and indirect paths (Figs 4 and 5).

**3.7.1. Pathway model of the determinants influencing life satisfaction among male respondents.** Fig 4 shows the pathway model to illustrate the relationship between the independent variables (demographic factors and year) and the dependent variables (life satisfaction) by sex (males) (Fig 4).

**3.7.2. Pathway model of the determinants influencing life satisfaction among female respondents.** Fig 5 below shows the pathway model to illustrate the relationship between the independent variables (demographic factors and year) and the dependent variables (life satisfaction) by sex (females) (Fig 5).

### 3.8. Generalized structural modelling (gSEM) analyses for male respondents (n = 33,325)

Tables 7 and 8 involves the generalized structural equation modelling (gSEM), which indicated the strength and direction of the relationships between the determinants and life satisfaction and the effects they have on each other, as direct, indirect and total effects.

**Table 7. Generalized structural equation modelling for the determinants of life satisfaction as a confirmed factor for the pooled samples of the male respondents (n = 33,325).**

| | | Direct effects | | | Indirect effects | | | | Total effects | | | |
|---|---|---|---|---|---|---|---|---|---|---|---|---|
| | | Coef. | P > \|Z\| | [95% Conf. Interval] | | Coef. | P > \|Z\| | [95% Conf. Interval] | | Coef. | P > \|Z\| | [95% Conf. Interval] | |
| **Effect of highest education on life satisfaction via year of survey** | | | | | | | | | | | | | |
| Primary | Suffering | Ref. | | | | | | | | | | | |
| | Struggling | 0.251 | 0.170 | −0.107 | 0.609 | 0.002 | 0.520 | −0.005 | 0.009 | 0.253 | 0.166 | −0.105 | 0.611 |
| | Thriving | 0.124 | 0.316 | −0.118 | 0.365 | 0.000 | 0.651 | −0.001 | 0.002 | 0.124 | 0.314 | −0.117 | 0.365 |
| Secondary | Suffering | Ref. | | | | | | | | | | | |
| | Struggling | 0.304 | 0.075 | −0.031 | 0.638 | −0.004 | 0.287 | −0.011 | 0.003 | 0.300 | 0.079 | −0.035 | 0.634 |
| | Thriving | 0.060 | 0.599 | −0.165 | 0.285 | −0.001 | 0.588 | −0.003 | 0.002 | 0.060 | 0.604 | −0.165 | 0.284 |
| Higher | Suffering | Ref. | | | | | | | | | | | |
| | Struggling | 0.553 | 0.002 | 0.199 | 0.908 | −0.005 | 0.204 | −0.013 | 0.003 | 0.548 | 0.002 | 0.193 | 0.903 |
| | Thriving | 0.493 | 0.000 | 0.254 | 0.732 | −0.001 | 0.571 | −0.004 | 0.002 | 0.492 | 0.000 | 0.253 | 0.731 |
| **Effect of employment on life satisfaction via year of survey** | | | | | | | | | | | | | |
| Employed | Suffering | Ref. | | | | | | | | | | | |
| | Struggling | 0.140 | 0.005 | 0.041 | 0.239 | 0.001 | 0.321 | −0.001 | 0.003 | 0.141 | 0.005 | 0.043 | 0.240 |
| | Thriving | 0.075 | 0.033 | 0.006 | 0.143 | 0.000 | 0.594 | −0.001 | 0.001 | 0.075 | 0.032 | 0.006 | 0.144 |
| **Effect of income on life satisfaction via year of survey** | | | | | | | | | | | | | |
| Low | Suffering | Ref. | | | | | | | | | | | |
| | Struggling | 0.436 | 0.000 | 0.267 | 0.604 | −0.029 | 0.030 | −0.055 | −0.003 | 0.407 | 0.000 | 0.238 | 0.575 |
| | Thriving | 0.480 | 0.000 | 0.367 | 0.594 | −0.005 | 0.543 | −0.023 | 0.012 | 0.475 | 0.000 | 0.362 | 0.588 |
| Middle | Suffering | Ref. | | | | | | | | | | | |
| | Struggling | 1.009 | 0.000 | 0.763 | 1.255 | −0.035 | 0.031 | −0.067 | −0.003 | 0.974 | 0.000 | 0.729 | 1.219 |
| | Thriving | 1.462 | 0.000 | 1.288 | 1.637 | −0.007 | 0.543 | −0.028 | 0.015 | 1.456 | 0.000 | 1.282 | 1.629 |
| High | Suffering | Ref. | | | | | | | | | | | |
| | Struggling | 0.953 | 0.000 | 0.659 | 1.247 | −0.038 | 0.031 | −0.073 | −0.003 | 0.915 | 0.000 | 0.622 | 1.207 |
| | Thriving | 1.875 | 0.000 | 1.658 | 2.091 | −0.007 | 0.543 | −0.030 | 0.016 | 1.868 | 0.000 | 1.652 | 2.083 |

Note: * p < 0 05, ** p < 0 01, *** p < 0 001.

**Table 8. Generalized structural equation modelling for the determinants of life satisfaction as a confirmed factor for the pooled samples of the female respondents (n = 36,989).**

| Demographic Determinants | | Direct effects | | | | Indirect effects | | | | Total effects | | | |
|---|---|---|---|---|---|---|---|---|---|---|---|---|---|
| | | Coef. | P>\|Z\| | [95% Conf. Interval] | | Coef. | P>\|Z\| | [95% Conf. Interval] | | Coef. | P>\|Z\| | [95% Conf. Interval] | |
| **Effect of highest education on life satisfaction via year of survey** | | | | | | | | | | | | | |
| Primary | Suffering | | | | | | | | | | | | |
| | Struggling | 0.224 | 0.078 | −0.025 | 0.472 | 0.002 | 0.321 | −0.002 | 0.006 | 0.226 | 0.075 | −0.023 | 0.474 |
| | Thriving | 0.309 | 0.000 | 0.140 | 0.477 | 0.006 | 0.105 | −0.001 | 0.012 | 0.314 | 0.000 | 0.145 | 0.484 |
| +<br>Secondary | Suffering | | | | | | | | | | | | |
| | Struggling | 0.139 | 0.238 | −0.091 | 0.368 | 0.006 | 0.237 | −0.004 | 0.016 | 0.144 | 0.219 | −0.086 | 0.374 |
| | Thriving | 0.131 | 0.096 | −0.023 | 0.285 | 0.017 | 0.000 | 0.008 | 0.026 | 0.148 | 0.062 | −0.008 | 0.303 |
| Higher | Suffering | | | | | | | | | | | | |
| | Struggling | 0.396 | 0.002 | 0.144 | 0.648 | 0.009 | 0.233 | −0.006 | 0.024 | 0.405 | 0.002 | 0.153 | 0.657 |
| | Thriving | 0.456 | 0.000 | 0.284 | 0.629 | 0.025 | 0.000 | 0.013 | 0.037 | 0.482 | 0.000 | 0.308 | 0.655 |
| **Effect of employment on life satisfaction via year of survey** | | | | | | | | | | | | | |
| Employed | Suffering | | | | | | | | | | | | |
| | Struggling | 0.156 | 0.001 | 0.067 | 0.245 | −0.002 | 0.251 | −0.004 | 0.001 | 0.155 | 0.001 | 0.066 | 0.243 |
| | Thriving | 0.011 | 0.733 | −0.054 | 0.077 | −0.005 | 0.003 | −0.008 | −0.002 | 0.007 | 0.836 | −0.058 | 0.072 |
| **Effect of income on life satisfaction via year of survey** | | | | | | | | | | | | | |
| Low | Suffering | | | | | | | | | | | | |
| | Struggling | 0.175 | 0.031 | 0.016 | 0.334 | 0.011 | 0.228 | −0.007 | 0.029 | 0.186 | 0.022 | 0.027 | 0.345 |
| | Thriving | 0.221 | 0.000 | 0.109 | 0.332 | 0.032 | 0.000 | 0.018 | 0.045 | 0.252 | 0.000 | 0.140 | 0.364 |
| Middle | Suffering | | | | | | | | | | | | |
| | Struggling | 0.664 | 0.000 | 0.422 | 0.906 | 0.014 | 0.229 | −0.009 | 0.036 | 0.678 | 0.000 | 0.436 | 0.920 |
| | Thriving | 1.295 | 0.000 | 1.121 | 1.469 | 0.039 | 0.000 | 0.022 | 0.055 | 1.334 | 0.000 | 1.159 | 1.508 |
| High | Suffering | | | | | | | | | | | | |
| | Struggling | 0.932 | 0.000 | 0.628 | 1.236 | 0.016 | 0.228 | −0.010 | 0.043 | 0.948 | 0.000 | 0.645 | 1.251 |
| | Thriving | 1.872 | 0.000 | 1.640 | 2.103 | 0.046 | 0.000 | 0.026 | 0.066 | 1.918 | 0.000 | 1.688 | 2.148 |

Note: * ρ<0 05, ** ρ<0 01, *** ρ<0 001.

**3.8.1. Generalized structural modelling (gSEM) analyses for male respondents (n = 33,325).** Table 7 presents the effects of education, employment, and income on life satisfaction (suffering, struggling, thriving). Our results show that higher education, employment, and income significantly enhance life satisfaction (p<0.000). Higher education directly influences life satisfaction, promoting struggling and thriving among males, while employment status notably increases life satisfaction for those struggling or thriving. Income levels (low, middle, high) also positively affect life satisfaction (p<0.000). Indirectly, higher education shapes employment status, which in turn influences life satisfaction (p<0.000). The combined direct and indirect effects highlight the comprehensive role of higher education, employment, and income in improving life satisfaction. Overall, employment status emerges as a key driver, directly boosting life satisfaction and amplifying the effects of education and income.

**3.8.2. Generalized structural modelling (gSEM) analyses for female respondents (n = 36,989).** Table 8 presents the effects of demographic factors—education, employment, and income—on life satisfaction (suffering, struggling, and thriving) among female respondents. Significant positive total effects were observed for struggling and thriving life satisfaction across higher education, employment status, and income levels (p ≤ 0.000). Education at all levels (primary, secondary, higher) directly increased struggling and thriving life satisfaction, as did employment and

income status (p ≤ 0.000). Indirectly, education influenced life satisfaction by improving employment status, which in turn raised income levels, both contributing to higher life satisfaction (p ≤ 0.000). The combined direct and indirect pathways highlight the comprehensive impact of education, employment, and income on life satisfaction among females.

### 3.8.3. Percentage distribution of GSEM mediation analysis of life satisfaction among male and female respondents.

Table 9 below shows SEM mediation percentages by gender, highlighting differences in life satisfaction determinants in Gauteng. Income was the strongest predictor, with high-income individuals reporting the highest total effects for both sexes. Thriving high-income women had a total effect of 1.918 (p < 0.001) with 2.4% mediation; men had slightly higher effects with similar mediation. Education positively influenced life satisfaction, especially at higher levels. Among thriving women, 5.2% of higher education's effect was mediated by survey year, compared to a lower rate for men. Secondary education for thriving women showed the highest mediation (11.5%), reflecting sensitivity to historical and policy shifts; men showed lower, more stable mediation. Employment effects differed by gender: for struggling women, employment increased life satisfaction (0.155, p = 0.001) with negligible mediation (−1.3%), while men had stronger direct and slightly higher indirect effects mediated by survey year. Overall, mediation was low (< 5%) except for secondary education among thriving women, indicating socio-economic factors primarily drive life satisfaction with modest gendered mediation by temporal and policy changes.

### 3.8.4. Key gendered findings.

Income is the strongest predictor of life satisfaction for both men and women, with only modest influence from historical or policy changes. Education positively affects life satisfaction across genders, but

**Table 9. GSEM Mediation Percentages for Life Satisfaction Determinants among Male (n = 33,325) and Female (n = 36,989) Respondents.**

| Demographic determinants | Males | | | | | | Females | | | | |
|---|---|---|---|---|---|---|---|---|---|---|---|
| | Category | Direct effects | Indirect effects | Total effect | % Mediated | Significance (Total) | Direct effects | Indirect effects | Total effect | % Mediated | Significance (Total) |
| Education (primary) | Suffering | RC | RC | RC | RC | RC | RC | RC | RC | RC | RC |
| | Struggling | 0.251 | 0.002 | 0.253 | 0.79% | 0.166 | 0.224 | 0.002 | 0.226 | 0.88% | 0.075 |
| | Thriving | 0.124 | −0.001 | 0.124 | −0.81% | 0.314 | 0.309 | 0.006 | 0.314 | 1.91% | 0.000 |
| Education (secondary) | Suffering | RC | RC | RC | RC | RC | RC | RC | RC | RC | RC |
| | Struggling | 0.304 | −0.004 | 0.300 | −1.33% | 0.079 | 0.139 | 0.006 | 0.144 | 4.17% | 0.219 |
| | Thriving | 0.060 | −0.003 | 0.060 | −5.00% | 0.604 | 0.131 | 0.017 | 0.148 | 11.9% | 0.062 |
| Education (higher) | Suffering | RC | RC | RC | RC | RC | RC | RC | RC | RC | RC |
| | Struggling | 0.553 | −0.005 | 0.548 | −0.91% | 0.002 | 0.396 | 0.009 | 0.405 | 2.22% | 0.002 |
| | Thriving | 0.493 | −0.004 | 0.492 | −0.81% | 0.000 | 0.456 | 0.025 | 0.482 | 5.19% | 0.000 |
| Employment (employed) | Suffering | RC | RC | RC | RC | RC | RC | RC | RC | RC | RC |
| | Struggling | 0.140 | −0.001 | 0.141 | −0.71% | 0.005 | 0.156 | −0.002 | 0.155 | −1.29% | 0.001 |
| | Thriving | 0.075 | −0.001 | 0.075 | −1.33% | 0.032 | 0.011 | −0.002 | 0.007 | −28.57% | 0.836 |
| Income (low) | Suffering | RC | RC | RC | RC | RC | RC | RC | RC | RC | RC |
| | Struggling | 0.436 | 0.030 | 0.407 | 7.37% | 0.000 | 0.175 | 0.007 | 0.186 | 3.76% | 0.022 |
| | Thriving | 0.480 | 0.012 | 0.475 | 2.53% | 0.000 | 0.221 | 0.018 | 0.252 | 7.14% | 0.000 |
| Income (middle) | Suffering | RC | RC | RC | RC | RC | RC | RC | RC | RC | RC |
| | Struggling | 1.009 | 0.035 | 0.974 | 3.59% | 0.000 | 0.664 | 0.014 | 0.678 | 2.06% | 0.000 |
| | Thriving | 1.462 | 0.015 | 1.456 | 1.03% | 0.000 | 1.295 | 0.039 | 1.334 | 2.92% | 0.000 |
| Income (higher) | Suffering | RC | RC | RC | RC | RC | RC | RC | RC | RC | RC |
| | Struggling | 0.953 | 0.031 | 0.915 | 3.39% | 0.000 | 0.932 | 0.016 | 0.948 | 1.69% | 0.000 |
| | Thriving | 1.875 | 0.016 | 1.868 | 0.86% | 0.000 | 1.872 | 0.066 | 1.918 | 3.44% | 0.000 |

Note: * p < 0 05, ** p < 0 01, *** p < 0 001.

secondary education among women shows the greatest responsiveness to socio-political shifts. Employment enhances life satisfaction for both sexes, with more stable and slightly better mediation for men; women's gains are most evident among vulnerable groups, showing little policy-driven change over time. Overall, mediation effects remain small for most socio-economic factors, though gender differences appear in education and employment.

## 4. Discussion

This study identified key determinants of life satisfaction among adults in Gauteng, South Africa, highlighting the influence of age, education, income, employment, household size, medical aid, and media access. While gender differences were modest, women generally reported higher satisfaction levels, shaped by cultural and socioeconomic factors. The findings support the U-curve hypothesis, with life satisfaction lowest in midlife and higher in early adulthood and after 48 years [66–68]. Regional variations persist, as life satisfaction declines with age in many low- and middle-income contexts due to economic and social instability [69–78]. These findings underscore the importance of addressing employment, education, and health access while considering gendered and age-specific needs to improve life satisfaction outcomes. Policymakers should implement age-targeted social programs, including financial literacy training, employment support, and healthcare access for middle-aged adults, and ensure social engagement and support systems for older adults to maintain high life satisfaction.

Career fulfilment, job satisfaction, financial stability, and concerns about retirement and debt significantly shape life satisfaction in middle age [70]. Supportive family and social relationships improve satisfaction, while conflict and poor work-life balance reduce it [76,79,80]. Personal development opportunities and a sense of purpose further enhance well-being [79,80]. For older adults, life satisfaction depends on health, financial security, social ties, living conditions, and healthcare access, with good health and economic stability promoting satisfaction and chronic illness or financial strain diminishing it [77,81]. A strong sense of community and accessible healthcare are also vital [77]. This study found significant associations between individual-, household-, and community-level factors and life satisfaction, underscoring their interconnected roles. Positive links between personal achievement, household stability, and community resources reinforce the importance of social cohesion. These findings support integrated policies addressing inequality, promoting equity, and advancing sustainable development goals [1,78]. Interventions targeting these key factors can meaningfully enhance life satisfaction across age groups. Governments and local authorities can promote workplace wellness programmes, career development initiatives, and retirement planning support. Community-based programmes that strengthen social networks, provide accessible healthcare, and improve housing conditions can enhance well-being across age groups.

Younger and older adults report higher life satisfaction than middle-aged individuals, reflecting differences in life stages, responsibilities, and stress levels [1,82]. Higher education improves life satisfaction by enhancing job prospects, income, and problem-solving abilities [82,83]. Cultural background and social identity also shape satisfaction through differing expectations and support systems [1,2]. Although higher income improves life satisfaction, its impact decreases once basic financial needs are met and access to essential resources is secured [1,24]. Employment contributes more to life satisfaction than unemployment by providing financial security, purpose, and social interaction [1,17]. In South Africa, key drivers of life satisfaction include higher income, stable employment, good health, education, supportive relationships, and low fear of crime [1,24]. Smaller households report greater satisfaction owing to reduced financial strain [84,85], and medical insurance is strongly associated with well-being [86,87]. Higher household income improves living conditions, healthcare, and education opportunities [57,87], while social grants help ease financial burdens in low-income households [1,24]. Access to media enhances life satisfaction by offering information, entertainment, and social connection [88,89]. Lastly, stable, quality housing contributes to security and life satisfaction, whereas poor housing diminishes it [1,90,91]. Targeted interventions should focus on improving access to quality education, equitable employment opportunities, and financial support for low-income households. Expanding affordable healthcare, social grants, and housing programmes

can reduce inequality and enhance well-being. Policies promoting safe neighborhoods and community engagement can further support life satisfaction.

Community-level factors such as safety, housing quality, media access, and public amenities play a crucial role in shaping life satisfaction for both men and women in South Africa [88–91]. Access to media fosters connection and reduces isolation, while safe, well-maintained housing offers stability and improves psychological well-being. Communities with strong social support networks, clean environments, and accessible facilities report higher life satisfaction [1,92]. In Gauteng, urban municipalities report greater life satisfaction due to better infrastructure, healthcare, education, and employment opportunities, while rural areas experience lower satisfaction due to limited services and resources [1,24]. These findings underscore the importance of physical living conditions, community infrastructure, and social cohesion in promoting life satisfaction across diverse settings. Employment opportunities and safety perceptions impact men's and women's life satisfaction differently. Urban areas offer diverse jobs, boosting life satisfaction, while rural areas have limited, gender-specific employment [1,24]. Higher urban crime rates particularly affect women's safety and satisfaction [92]. Stronger community ties and traditional gender roles in rural areas influence life satisfaction differently by gender [93–95]. Tailored policies addressing the distinct needs of men and women in urban and rural settings, improving infrastructure, services, and safety can enhance overall life satisfaction. Urban and rural planning should prioritize equitable infrastructure development, improve public transportation, enhance safety (particularly for women), and expand access to healthcare, education, and media. Programs promoting rural employment opportunities, skills training, and gender-sensitive job placement can reduce disparities.

This study used generalized structural equation modeling (gSEM) to examine how education, employment, and income mediate the relationship between sex and life satisfaction (struggling and thriving). For males, primary to higher education, employment, and income significantly influenced struggling life satisfaction, with higher education, employment, and income boosting thriving life satisfaction consistently across survey years [96–102]. Stable, well-paying jobs and job security were linked to better male life satisfaction. Similarly, these determinants influenced female life satisfaction throughout the survey period, with higher education and income promoting thriving life satisfaction [99,100,103–122]. Education enhances job opportunities, income, and personal achievement, while income reduces stress and offers fulfilling experiences [1,57,89,111]. Employment provides purpose and financial independence, with job quality and work-life balance being key [89,112]. Women's physical and mental health, along with strong social networks, are crucial for life satisfaction, offering emotional support and belonging [69,99,103,110]. These findings align with prior research showing that demographic factors foster life satisfaction for both sexes, moderated by individual and cultural differences [1,24]. Gender-sensitive policies should focus on improving women's access to quality education, fair employment, and income-generating opportunities. Workplace policies addressing job quality, flexible schedules, and work-life balance can enhance life satisfaction for both sexes.

Female-headed households often face lower quality of life and greater economic vulnerability than male-headed ones, partly due to commuting challenges that limit employment access [89–94,97,101]. These households rely more on social grants, reflecting gender disparities in urbanization and intersecting with race and ethnicity to affect resource access and well-being [90,91,94]. Programmes targeting female-headed households, such as subsidized childcare, transportation support, skills training, and income support, can reduce economic vulnerability and promote equity. Urban planning should improve accessibility to essential services and employment opportunities. Addressing these inequalities requires targeted policies to support female-headed households, often single-parent or skipped-generation families by enhancing earning capacities and reducing grant dependency [90,98,103,106]. Mapping data from the GCRO (2009–2024) can guide transportation and urban planning to improve accessibility, especially for women, informing strategies to promote gender equality and service access in Gauteng [28–34,98,103]. Income remains the strongest determinant of life satisfaction for both genders, with limited mediation over time, highlighting persistent socio-economic inequalities shaping well-being in South Africa.

Education's impact on life satisfaction shows clear gender differences. While higher education improves life satisfaction for both sexes, women especially with secondary education are more affected by policy and temporal changes, indicating middle-tier education as a key intervention point [123–125]. Employment effects also differ as men experience more stable and mediated benefits over time, whereas women, particularly those struggling, see direct but less mediated gains. Policies should promote gender-responsive education programmes, support women's progression beyond secondary education, and address structural labour market barriers. Employment programmes should consider gender-specific needs, including flexible work arrangements, mentorship, and skills development for women [125,126]. Income remains the strongest determinant of life satisfaction for both genders, highlighting persistent socio-economic inequalities shaping well-being. This reflects persistent structural barriers in the labour market for women, where employment alone may not improve quality of life without gender-responsive policies [127,128]. These findings call for gender-sensitive approaches that address the evolving and complex socio-economic inequalities beyond income and education. Broader socio-economic policies addressing income inequality, social protection, and access to essential services are crucial for improving overall life satisfaction across Gauteng.

## 4.1. Further discussion: Implementation science for gender-sensitive psychodemographic and sociology practices

Gender sensitivity promotes respect across sexes, helping to avoid stereotypes and benefit all [113,114]. Gender-transformative interventions challenge harmful norms, especially around masculinity, and are vital to addressing complex inequalities in resources, power, and opportunities [115]. Effective gender-sensitive policies must address specific needs to promote equality and life satisfaction. Implementation science plays a crucial role by translating research into practice, identifying effective interventions, overcoming barriers, and enabling continuous monitoring and adjustment [112,115–117]. Engaging stakeholders ensures relevance and acceptance, supporting sustainable, impactful gender equality initiatives [101,118]. South Africa's gender equality efforts, including the National Policy Framework and UN Women's strategy [104,108], seek to close gender gaps, but disparities persist. Gender-sensitive policies that ensure equal access to resources, opportunities, and protections can improve life satisfaction and promote equity [114,115]. Implementation science is essential to making these policies effective and advancing outcomes for all genders [115–118]. Raising awareness and promoting gender equality further enhances life expectations and social well-being.

## 4.2. Strengths and limitations

This study's key strength lies in its gender-specific analysis of life satisfaction determinants within Gauteng's social context, providing nuanced insights into male and female experiences. Utilizing the provincially representative GCRO QoL dataset (2009–2024) enhances reliability and generalizability while minimizing self-report biases. Integrating perspectives from sociology, psychology, and demography further deepens understanding of how individual, household, and community factors shape life satisfaction and highlight areas for improvement. However, the cross-sectional design limits causal inference and trend analysis. Future research should adopt longitudinal approaches and advanced methods like multilevel modeling to better capture gender-specific variations over time. The omission of variables such as marital status and religion due to non-response may affect model accuracy, so including these in future surveys is recommended. Overall, this study expands the literature on life satisfaction in Gauteng, offering a comprehensive foundation for future gender-focused research.

## 4.3. Implications for sociology of life satisfaction and pyschodemographic research

Our research shows that social factors mediate the relationship between demographic determinants and life satisfaction. Understanding these determinants aids in creating targeted interventions to improve life satisfaction in South Africa. Gender equality plays a crucial role by challenging traditional roles, improving resource access, and promoting fairness, which correlates with higher life satisfaction for both men and women [70,110]. Gender equality empowers women's

decision-making and reduces pressures on men to conform, fostering diverse interests and careers. Psychodemographic and sociological research should guide gender-sensitive interventions that address the distinct needs of both sexes. Implementing evidence-based policies and programmes based on these insights can build a more equitable society where everyone can achieve greater well-being.

## 5. Conclusions

Our study examined how demographic determinants and gender influence life satisfaction among men and women. We found strong links between education, employment, income, and gendered experiences of struggling or thriving life satisfaction. Gender moderated the association between municipalities and suffering-related life satisfaction, reflecting how lived experiences and social roles vary by location. Including gender in life satisfaction research is vital, as it shapes career, family, and identity outcomes. Promoting gender equality and challenging stereotypes fosters a society with equal rights and opportunities for all [100,101]. Gender-responsive programmes enhance well-being across genders [100,101]. Although socio-economic factors directly affect life satisfaction in Gauteng, the impact of historical and policy changes remains modest, especially for women. These results highlight the need for targeted, gender-sensitive policies that address structural inequalities and strengthen public policy's mediating role. Based on these findings, we propose three key recommendations for South Africa:

### Research and data collection

The Gauteng City Region Observatory (GCRO) should integrate variables from sociology, psychology, and psychodemographics into its surveys to enable intersectional analyses of gendered life satisfaction and well-being. This will enhance the understanding of how gender intersects with socio-economic and spatial inequalities, informing more targeted, inclusive policy responses.

### Targeted intervention and programme initiatives

Municipalities facing economic and social decline need inclusive economic empowerment initiatives for both women and men. Concurrently, public sensitisation campaigns on gender roles and stereotypes should promote equality, enabling individuals to pursue personal and professional goals free from traditional constraints.

### Policy advocacy

Provinces must adopt comprehensive, gender-responsive policy reforms addressing systemic inequalities. Strengthening and funding gender advocacy networks is essential to ensure sustained support for the diverse needs of women and men and to advance equitable socio-economic development.

## Supporting information

**S1 File. Do files as stata file.**
(ZIP)

## Acknowledgments

The authors extend their sincere gratitude to the Gauteng City-Region Observatory (GCRO) for granting access to the survey dataset, which was instrumental in conducting this research. Monica Ewomazino Akokuwebe acknowledges her affiliation with the Department of Science and Innovation and the National Research Foundation Centre of Excellence in

Human Development at the University of the Witwatersrand, Johannesburg, South Africa. The views expressed in this study are solely those of the authors and do not necessarily reflect the perspectives of the GCRO or any affiliated institutions. Finally, we would like to thank Mrs. Helen Thomas for her meticulous support in language editing, which significantly enhanced the clarity and precision of this work.

## Author contributions

**Conceptualization:** Monica Ewomazino Akokuwebe.

**Data curation:** Salmon Likoko, Shamsunisaa Miles-Timotheus.

**Formal analysis:** Monica Ewomazino Akokuwebe, Salmon Likoko, Shamsunisaa Miles-Timotheus, Moyahabo Mabala.

**Investigation:** Monica Ewomazino Akokuwebe.

**Methodology:** Monica Ewomazino Akokuwebe, Salmon Likoko.

**Project administration:** Monica Ewomazino Akokuwebe.

**Resources:** Monica Ewomazino Akokuwebe.

**Software:** Salmon Likoko, Shamsunisaa Miles-Timotheus.

**Supervision:** Monica Ewomazino Akokuwebe.

**Validation:** Monica Ewomazino Akokuwebe, Salmon Likoko, Shamsunisaa Miles-Timotheus, Moyahabo Mabala.

**Visualization:** Monica Ewomazino Akokuwebe, Shamsunisaa Miles-Timotheus, Moyahabo Mabala.

**Writing – original draft:** Monica Ewomazino Akokuwebe.

**Writing – review & editing:** Monica Ewomazino Akokuwebe, Salmon Likoko.

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
