## [Decision Letter · Decision Letter 0]

12 Aug 2025

PONE-D-25-37543“Gender Dimensions of Quality of Life”: The Determinants of Life Satisfaction Among South Africans Residing in the Gauteng Province—Using a Multilevel Psychodemographic Analysis of the GCRO’s Quality of Life Survey (2009–2024)PLOS ONE

Dear Dr. Akokuwebe,

Thank you for submitting your manuscript to PLOS ONE. After careful consideration, we feel that it has merit but does not fully meet PLOS ONE’s publication criteria as it currently stands. Therefore, we invite you to submit a revised version of the manuscript that addresses the points raised during the review process.

We look forward to receiving your revised manuscript.

Kind regards,

Dr Siyabulela Christopher Fobosi

Academic Editor

PLOS ONE

Journal Requirements:

3.We note that Figure(s) 4 and 5 in your submission contain [map/satellite] images which may be copyrighted. All PLOS content is published under the Creative Commons Attribution License (CC BY 4.0), which means that the manuscript, images, and Supporting Information files will be freely available online, and any third party is permitted to access, download, copy, distribute, and use these materials in any way, even commercially, with proper attribution. For these reasons, we cannot publish previously copyrighted maps or satellite images created using proprietary data, such as Google software (Google Maps, Street View, and Earth). For more information, see our copyright guidelines: http://journals.plos.org/plosone/s/licenses-and-copyright.

a. You may seek permission from the original copyright holder of Figure(s) 4 and 5 to publish the content specifically under the CC BY 4.0 license.

5. Please remove your figures from within your manuscript file, leaving only the individual TIFF/EPS image files, uploaded separately. These will be automatically included in the reviewers’ PDF**.**

6. We are unable to open your Supporting Information file [Do files as stata file.zip]. Please kindly revise as necessary and re-upload.

Reviewers' comments:

Reviewer's Responses to Questions

**Comments to the Author**

1. Is the manuscript technically sound, and do the data support the conclusions?

Reviewer #1: Yes

2. Has the statistical analysis been performed appropriately and rigorously? 

Reviewer #1: Yes

3. Have the authors made all data underlying the findings in their manuscript fully available?

Reviewer #1: Yes

4. Is the manuscript presented in an intelligible fashion and written in standard English?

Reviewer #1: Yes

5. Review Comments to the Author

Reviewer #1: Paper is technically sound. Secondary data has been properly utilized and proper statistical analysis was done. paper was written in good English and also could'nt find any grammatical mistakes. Authors really have put effort in the paper.

6. PLOS authors have the option to publish the peer review history of their article (what does this mean? ). If published, this will include your full peer review and any attached files.

**Do you want your identity to be public for this peer review?** For information about this choice, including consent withdrawal, please see our Privacy Policy .

Reviewer #1: No

---

## [Author Response · Author response to Decision Letter 1]

1 Sep 2025

Rebuttal letter

• A rebuttal letter that responds to each point raised by the academic editor and reviewer(s). You should upload this letter as a separate file labelled 'Response to Reviewers'.

Authors’ Response: We have prepared the rebuttal letter that responds to each point raised by the academic editor and the reviewers. The letter has been uploaded as a separate file labelled “Response to Reviewers.” It presents each comment in full, followed by our point by point reply, and specifies the exact page and line numbers where revisions were made in the manuscript.

The letter also

• summarizes substantive changes at the beginning for quick orientation

• explains any new analyses, clarifications, or updates to figures, and Supporting Information

• notes where in text citations and reference list entries were adjusted

• indicates cases where a suggestion did not lead to a change and provides a brief rationale

All revisions in the manuscript are clearly marked for ease of review, and the clean version reflects the final text. The format of the letter follows PLOS ONE guidance.

• A marked-up copy of your manuscript that highlights changes made to the original version. You should upload this as a separate file labelled 'Revised Manuscript with Track Changes'.

Authors’ Response: We have prepared a marked-up copy of the manuscript that highlights all changes made to the original version. This document has been formatted in accordance with the journal’s requirements and will be uploaded as a separate file labelled “Revised Manuscript with Track Changes.” The tracked changes clearly show insertions, deletions, and modifications throughout the manuscript for ease of review.

• An unmarked version of your revised paper without tracked changes. You should upload this as a separate file labelled 'Manuscript'.

Authors’ Response: We have also prepared a clean, unmarked version of the revised manuscript without tracked changes. This file will be uploaded separately and labelled “Manuscript.” It reflects all the revisions made and represents the final version for consideration.

Journal Requirements

1a. When submitting your revision, we need you to address these additional requirements. Please ensure that your manuscript meets PLOS ONE's style requirements, including those for file naming. The PLOS ONE style templates can be found at https://journals.plos.org/plosone/s/file?id=wjVg/PLOSOne _formatting_sample_main_body.pdf

Authors’ Response: We have addressed these additional requirements by ensuring that our manuscript meets the PLoS ONE’s style requirements, including those for filing naming. We followed the PLoS One Manuscript Body Formatting Guidelines - Abstract, Introduction, Materials and Methods, Results, Discussion, Conclusions, Acknowledgements, References, Supporting Information. [Page 1-Page 34]

1b. and https://journals.plos.org/plosone/s/file?id=ba62/PLOSOne_format ting_sample_title_authors_affiliations.pdf

Authors’ Response: We have addressed the comments to ensure that the Authors’ names, Authors’ Affiliations, and Corresponding Author Email has been fixed properly in line with the guidelines of the PLoS ONE Journal Requirement Formatting Guidelines. [Page 1]

Authors’ Response: Data Availability Statement: All relevant data underlying the findings of this study will be made available in a public repository upon acceptance of the manuscript. De-identified data will be deposited in [insert repository name, e.g., Dryad, Figshare, Zendo, or institutional repository] and a DOI link will be provided in the final published article. Any potentially identifying or sensitive participant information will be removed to ensure compliance with ethical approval and participant confidentiality. [Page 27]

3. We note that Figure(s) 4 and 5 in your submission contain [map/satellite] images which may be copyrighted. All PLOS content is published under the Creative Commons Attribution License (CC BY 4.0), which means that the manuscript, images, and Supporting Information files will be freely available online, and any third party is permitted to access, download, copy, distribute, and use these materials in any way, even commercially, with proper attribution. For these reasons, we cannot publish previously copyrighted maps or satellite images created using proprietary data, such as Google software (Google Maps, Street View, and Earth). For more information, see our copyright guidelines: http://journals.plos.org/plosone/s/licenses-and-copyright.

a. You may seek permission from the original copyright holder of Figure(s) 4 and 5 to publish the content specifically under the CC BY 4.0 license.

We recommend that you contact the original copyright holder with the Content Permission Form (http://journals.plos.org/plosone/s/file?id=7c09 /content-permission-form. pdf) and the following text:

The following resources for replacing copyrighted map figures may be helpful: USGS National Map Viewer (public domain): http://viewer.nationalmap.gov/ viewer/

The Gateway to Astronaut Photography of Earth (public domain): http://eol.jsc.nasa.gov/ sseop/clickmap/

Maps at the CIA (public domain): https://www.cia.gov/library/publications/ the-world-factbook/index.html

and https://www.cia.gov/library/publications/cia-mapspublications/index. html

Authors’ Response: We would like to clarify that Figures 4 and 5 are original creations by the authors and are not subject to third-party copyright restrictions. These figures are choropleth maps generated using the freely available Datawrapper tool. Datawrapper enables users to design and export graphics while retaining full copyright over the outputs. The small footer (“Created with Datawrapper”) visible on the maps is simply a tool attribution and does not imply ownership or copyright by Datawrapper or any other third party. Importantly, the figures do not incorporate or reproduce any proprietary or copyrighted map or satellite imagery (e.g., Google Maps, Google Earth, or Street View).

The underlying data used to create these maps are from the Gauteng City-Region Observatory (GCRO) Quality of Life (QoL) surveys. These datasets are publicly available under a CC BY-SA 4.0 license via the GCRO Data Catalogue, and they are the same data that support the other analyses presented in the manuscript. This ensures compliance with open access and data-sharing requirements.

For clarity, we have revised the manuscript text under the sub-heading 2.9 Data Preparation and Analysis (lines 271–272) to explicitly state that the maps are original creations by the authors. In addition, the captions for Figures 4 and 5 have been updated to include author attribution and the source of the underlying data. These revisions appear under the relevant sub-sections of the Results: 3.6 Choropleth Maps Illustrating the Spatial Distribution of Life Satisfaction (Suffering) Stratified by Sex in Gauteng Municipalities; 3.6.1 Choropleth Maps Showing Distribution of Suffering Life Satisfaction among Males (n = 33,325); and 3.6.2 Choropleth Maps Showing Distribution of Suffering Life Satisfaction among Females (n = 36,989).

The updated figures and captions are included in the revised submission, with the changes highlighted in red for ease of reference.

Authors’ Response:

We confirm that a separate caption has been provided for each figure in the manuscript. These captions were previously included in blue ink in the main text to clearly distinguish them during the revision process. For reference, the captions appear in the manuscript as follows:

• Figure 1: Lines 311–313

• Figure 2: Lines 321–324

• Figure 3: Lines 334–337

• Figure 4: Page 16 – Distribution of life satisfaction by suffering among males in Gauteng

• Figure 5: Page 16 – Suffering among males in Gauteng province (2009–2024)

• Figure 6: Page 19 – Structural equation model for demographic factors

• Figure 7: Page 19 – Structural equation model for demographic factors

We have carefully reviewed the manuscript to ensure that each figure is accompanied by a distinct caption. All figure captions have been included in the revised submission and formatted according to PLOS ONE requirements for clarity and ease of reference.

5. Please remove your figures from within your manuscript file, leaving only the individual TIFF/EPS image files, uploaded separately. These will be automatically included in the reviewers’ PDF.

Authors’ Response:

We have followed the journal’s instructions and removed all figures from the main manuscript file. Each figure has been uploaded separately as an individual TIFF/EPS file in accordance with PLOS ONE submission guidelines. These files have been correctly labelled and formatted to ensure they are automatically included in the reviewers’ PDF. This adjustment has been carefully checked to maintain consistency between the manuscript text and the corresponding figures.

6. We are unable to open your Supporting Information file [Do files as Stata file.zip]. Please kindly revise as necessary and re-upload.

Authors’ Response:

We have revised the Supporting Information file to ensure that the Do-files can be accessed and executed correctly. To open the STATA datasets using the Do-files, please follow these steps:

1. Open STATA on your computer.

2. Drag and drop the datasets into the STATA window. A file path link will appear.

3. Copy this link and paste it at the beginning of the corresponding Do-file.

4. Once the link is correctly inserted, you can run the Do-file, and the datasets will open automatically.

For instance, on our system, the path appears as:

C:\Users\profm\OneDrive\Desktop\2025_PLOS One GCRO Submission \1_2025_PLoS Global Public Health\Do file Data and MS Word\ Datasets\GCRO 2011_28Feb12noGIS.dta

We have included this instruction in the revised Supporting Information file to guide reviewers in accessing and executing the Do-files without difficulty.

7. Please include captions for your Supporting Information files at the end of your manuscript, and update any in-text citations to match accordingly. Please see our Supporting Information guidelines for more information: http://journals.plos. org/plosone/s/supporting-information

Authors’ Response:

We have revised the manuscript to ensure compliance with the PLOS ONE Supporting Information guidelines. Specifically, captions for all Supporting Information files have now been included at the end of the manuscript. Each caption clearly describes the content of the respective file to assist readers in understanding the purpose and use of the material provided.

In addition, all in-text citations referencing the Supporting Information files have been updated to match the revised captions. This ensures consistency between the main manuscript text and the Supporting Information section. We have carefully checked the formatting and placement of these citations to align with PLOS ONE requirements.

The updated manuscript, with these revisions highlighted, has been resubmitted for review. See below the Supplementary Datasets and Supporting Information.

Supplementary Datasets:

S1. gcro-qlf-2009-v1.1-20011209-s12. This is the S1 GCRO dataset

S2. GCRO 2011_28Feb12noGIS. This is the S2 2011 GCRO dataset

S3. qol-iii-2013-2014-v1. This is the S3 2013 GCRO dataset

S4. qol-iv-2014-2015-v1. This is the S4 2014-2015 GCRO dataset

S5. qols-v-2017-2018-v2. This is the S4 2017-2018 GCRO dataset

S6. qols-2020-2021-new-weights-v1. This is the S4 2020-2021 GCRO dataset

S7. qols-2023-2024-weights-v2 (Stata). This is the S4 2023-2024 GCRO dataset

Supporting information:

S1 Fig. 1. [This is the S1 Fig legend - Figure 1. Sex distribution of respondents. The vertical bar chart depicts the percentage distribution of male and female respondents residing in Gauteng, South Africa. The chart illustrates the percentage distribution of observations for each data point or the grouping of the data points of sex by survey years (2009–2024)]

S2 Fig. 2. [This is the S2 Fig legend - Figure 2. Distribution of life satisfaction levels—suffering, struggling, and thriving—by sex. The grouped bar chart shows that the majority of both male and female respondents reported thriving life satisfaction in Gauteng Province, South Africa. The percentages were calculated from a representative sample, with life satisfaction levels stratified by sex (male and female)]

S3 Fig. 3. [This is the S2 Fig legend - Figure 3. Shows the life satisfaction levels by age as reported by the respondents who are South Africans, residing in Gauteng province, South Africa. The percentage was determined from the representative sample of the number of life satisfaction scales distributed by the different ages across the survey years (2009–2024)]

S4 Fig. 4. [This is the S4 Fig legend – Figure 4. Distribution of life satisfaction by suffering among males in Gauteng province (2009–2024)]

S5 Fig. 5. [This is the S5 Fig legend – Figure 5. Distribution of life satisfaction by suffering among females in Gauteng province (2009–2024)]

S6 Fig. 6. [This is the S6 Fig legend – Figure 6. Structural equation model for demographic factors (education, employment, and income), and life satisfaction (struggling and thriving) by sex (males) across the survey years. * p < 0.005; ** p ≤ 0.001.

S7 Fig. 7. [This is the S7 Fig le

---

## [Editor Report · Decision Letter 1]

3 Sep 2025

“Gender Dimensions of Quality of Life”: The Determinants of Life Satisfaction Among South Africans Residing in the Gauteng Province—Using a Multilevel Psychodemographic Analysis of the GCRO’s Quality of Life Survey (2009–2024)

PONE-D-25-37543R1

Dear author,

We’re pleased to inform you that your manuscript has been judged scientifically suitable for publication and will be formally accepted for publication once it meets all outstanding technical requirements.

Kind regards,

Siyabulela Christopher Fobosi

Academic Editor

PLOS ONE
---

## [Editor Report · Acceptance letter]

PONE-D-25-37543R1

PLOS ONE

Dear Dr. Akokuwebe,

I'm pleased to inform you that your manuscript has been deemed suitable for publication in PLOS ONE. Congratulations! Your manuscript is now being handed over to our production team.

Kind regards,

on behalf of

Dr. Siyabulela Christopher Fobosi

Academic Editor

PLOS ONE